# A Geometric Unification of Generative AI with Manifold-Probabilistic Projection Models

## Abstract

Most models of generative AI for images assume that images are inherently low-dimensional objects embedded within a high-dimensional space. Additionally, it is often implicitly assumed that thematic image datasets form smooth or piecewise smooth manifolds. Common approaches overlook the geometric structure and focus solely on probabilistic methods, approximating the probability distribution through universal approximation techniques such as the kernel method. In some generative models the low dimensional nature of the data manifest itself by the introduction of a lower dimensional latent space. Yet, the probability distribution in the latent or the manifold's coordinate space is considered uninteresting and is predefined or considered uniform. This study unifies the geometric and probabilistic perspectives by providing a geometric framework and a kernel-based probabilistic method simultaneously. The resulting framework demystifies diffusion models by interpreting them as a projection mechanism onto the manifold of "good images". This interpretation leads to the construction of a new deterministic model, the Manifold-Probabilistic Projection Model (MPPM), which operates in both the representation (pixel) space and the latent space. We demonstrate that the Latent MPPM (LMPPM) outperforms the Latent Diffusion Model (LDM) across various datasets, achieving superior results in terms of image restoration and generation.

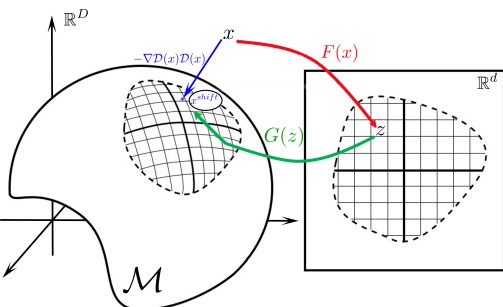

Figure 1: Illustration of our manifold-aware restoration approach. The blue path shows direct projection onto manifold $\mathcal{M}$ using distance function $\mathcal{D}_{\mathcal{M}}(x)$, while the red-green path represents encoding-decoding through latent space $\mathbb{R}^d$ via functions $F$ and $G$. Ideally, both paths converge to the same manifold point, ensuring geometrically consistent restoration.

## 1 Introduction

Restoration of images refers to the inverse process of generating a clean, meaningful, and non-corrupted image from a noisy, blurred, or other degraded input. A critical aspect of this process involves the use of prior knowledge or a well-approximated distribution function

over the set of clean images within a specific class. In this work, we propose the manifold assumption, which asserts that the set of desired images resides on a low-dimensional smooth manifold. We integrate this assumption with a probabilistic perspective. Specifically, we extend the conventional Monge patch description of the data manifold, typically provided by generative models such as autoencoders (AE) (Rumelhart & McClelland, 1987), variational autoencoders (VAE) (Kingma & Welling, 2013), and generative adversarial networks (GAN) (Goodfellow et al., 2014). Our approach augments this description by introducing a distance function that assigns, for each point in the pixel (ambient/representation) space, the distance to the closest point on the manifold. We treat here images as primary examples, but evidently it can be applied to any dataset that has this manifold structure. Next, we establish a connection between the geometric framework and the probabilistic perspective by introducing a geometric-based probability function and its kernel-based approximation. We further relate these approaches to diffusion-like methods, utilizing the score function to generate, in the ambient space, a vector field that directs each noisy or corrupted image towards the closest point on the manifold of clean images. By iteratively following this vector field, a diffusion-like flow is generated, guiding the corrupted image progressively towards a clean image residing on the manifold.

To accommodate the possibility of a nonuniform probability distribution on the manifold, we employ a kernel method that adjusts the diffusion-like flow to balance the trade-off between proximity to the manifold and the probability of a point on the manifold representing a clean and meaningful image. This integration of geometric principles with the kernel method constitutes the primary novelty of our approach. Furthermore, we extend these general concepts, the distance function, score, and diffusion-like flow, to operate within the latent space, thereby reducing computational complexity and enhancing the accuracy of the distance function. We evaluated our proposed method on the MNIST, SCUT-FBP5500 and CelebA-HQ-256 datasets, demonstrating superior performance compared to a leading method such as the Latent Diffusion Model (LDM) (Rombach et al., 2022).

## 1.1 Related Work

In recent years, the task of generating samples from a distribution that characterizes a specific dataset or target image has emerged as a critical challenge in machine learning. This problem has been extensively studied, with solutions primarily leveraging neural networks within deep learning frameworks. Many contemporary generative models operate under the implicit assumption that datasets comprise low-dimensional objects embedded within a high-dimensional space. However, the underlying geometry of the dataset is not always explicitly considered. For instance, variational autoencoders (VAEs) (Kingma & Welling, 2013) and Generative Adversarial Networks (GANs) (Goodfellow et al., 2014) construct a functional mapping from the low-dimensional latent space to the high-dimensional pixel space. This functional mapping can be interpreted as a transformation from the manifold coordinate system to the pixel coordinate system. More recent approaches, such as diffusion models (Sohl-Dickstein et al., 2015; Ho et al., 2020a), adopt a more implicit perspective on manifold structure. Geometrically, these models can be viewed as learning a directional field that guides noisy points back to the data manifold, enabling iterative projection. The diffusion process gradually transforms random noise into realistic samples by iteratively denoising along paths that converge onto the data manifold.

A central concept in many of these generative approaches is the Manifold Hypothesis (Loaiza-Ganem et al., 2024), which posits that real-world high-dimensional data, such as images, often concentrates near a low-dimensional manifold embedded within the ambient space. This geometric perspective provides a powerful conceptual framework for understanding generative models and has significantly influenced the design of numerous architectures and training objectives. Various other manifold-aware generative approaches have been proposed. Riemannian flow models (Gemici et al., 2016; Mathieu & Nickel, 2020) incorporate Riemannian metrics into flow-based models to explicitly account for the intrinsic geometry of the data manifold. The relationship between manifold structure and probabilistic frameworks remains an active area of research. Normalizing flows (Rezende & Mohamed, 2015) can be interpreted as learning diffeomorphisms between the data manifold and a simple base distribution. Score-based generative models (Song & Ermon, 2020) utilize the score

function (the gradient of the log-density) to characterize the data distribution, establishing a direct connection to the geometry of the data manifold. Recent works on denoising diffusion models (Ho et al., 2020b) can also be interpreted as learning a vector field that guides noisy samples back to the data manifold. Despite these advancements, there remains a gap in unifying the geometric and probabilistic perspectives in generative modeling.

This work addresses this gap by providing a geometric interpretation of autoencoders, leveraging geometric properties of the data, specifically the distance function to the manifold. We propose a new generative model that synthesizes both geometric and probabilistic approaches, leading to improved performance in generating high-quality samples. Our approach is based on the premise that the data manifold can be represented as a low-dimensional submanifold embedded within a high-dimensional space. We simultaneously learn both the distance function to this manifold and the probability distribution on it.

## 2 Background and Theoretical Framework

Many generative networks assume that images lie on a lower-dimensional manifold defined according to the latent space representation, which is embedded within a higher-dimensional representation space, such as the pixel space or ambient space. This manifold is explicitly modeled by the decoder in autoencoders (AEs) and variational autoencoders (VAEs), and by the generator in various Generative Adversarial Network (GAN) architectures. In all of these models, the manifold $\mathcal{M}$ is represented as a Monge patch. Let the latent space be $d$-dimensional, parameterized by $z$, and the pixel space be $D$-dimensional, parameterized by $x$, that is (see Fig. 1):

$$G(z) = \big(x_1(z_1, \ldots, z_d), \ldots, x_D(z_1, \ldots, z_d)\big).$$

In simple terms, the value at each pixel in the image (or in similar manifold-structured data) is a function of the $d$ parameters $z$. Many works, in the context of deep learning, use this representation to analyze the data set as a Riemannian manifold. We will mention here, as examples, (Shao et al., 2018; Wang & Ponce, 2021) where geodesics and directions of meaningful changes on the manifold are studied. In (Chadebec & Allassonnière, 2022) the relation of the induced metric of the manifold was found to be approximated close enough to the encoded point of a clean image by the inverse covariance found in VAE.

Another (implicit) way to describe a manifold is as the zero level set of a function. The distance function to the manifold in the ambient (representation) space is well suited for this purpose and is defined as follows:

$$\mathcal{D}_{\mathcal{M}}(x) = \min_{y \in \mathcal{M}} \|x - y\|, \tag{1}$$

where $\| \cdot \|$ denotes the Euclidean norm. In this high-dimensional representation space, the distance function provides a natural measure of the proximity of a point to the manifold. It is well known that $\mathcal{D}_{\mathcal{M}}$ satisfies the Eikonal equation (Hamilton, 1828) $\|\nabla \mathcal{D}_{\mathcal{M}}(x)\| = 1$, with the natural boundary condition $\mathcal{D}_{\mathcal{M}}(x) = 0$ for all $x \in \mathcal{M}$. Moreover, it is clear that $-\nabla \mathcal{D}_{\mathcal{M}}(x)$ defines a vector field pointing in the direction of the shortest path to the manifold.

Building on this purely geometric consideration, we introduce a probabilistic model. Following works such as (Kadkhodaie et al., 2023)(Sun et al., 2025). We start by assuming some non-trivial distribution of clean data $P_c(x)$ from which we have many samples, i.e. our data set. The probability of a non-data point $x$ is defined such that the resulting score vector field points toward the data manifold and its more densely populated regions. We therefore choose naturally the conditional probability of the corrupted image $x$ conditioned on clean data point $x'$ as $P(x|x') = f(\mathcal{D}(x, x'))$ where $\mathcal{D}(x, x')$ is the distance between the clean and corrupted image and $f$ is a monotonically decreasing function. For ease of analysis and computation we choose $f$ to be a Gaussian. The second assumption is the standard one, $\mathcal{D}(x, x') = \|x - x'\|$. These considerations lead to the following expression for the conditional probability function

$$P_\sigma(x|x') = \frac{1}{Q_d} \exp\left(-\frac{\|x - x'\|^2}{2\sigma^2}\right), \tag{2}$$

and the probability on the ambient space is then the well-known expression (Kadkhodaie et al., 2023)(Sun et al., 2025)

$$P(x;\sigma) = \int_{\mathbb{R}^D} P_\sigma(x|x')P_c(x')dx' = \frac{1}{Q_d} \int_{\mathbb{R}^D} \exp\left(-\frac{\|x-x'\|^2}{2\sigma^2}\right)P_c(x')dx', \qquad (3)$$

where $P_c(x')$ stands for the probability of the clean image $x'$. Although the Gaussian form might suggest a restriction to Gaussian noise, this formulation imposes no specific assumption regarding the degradation process that transforms $x'$ to $x$. It only assumes that the likelihood decreases exponentially with the distance between the corrupted and clean images. The result is a blind image denoising that does not need to have the type of noise or its amplitude as input.

Calculating the score of this probability function is impossible because of the need to find the mean over the whole ambient space. One of our main contributions is to introduce in this formulation the manifold hypothesis, namely we propose that $P_c(x') = \int P((G(z))\delta(x' - G(z))dV_\mathcal{M}$. This means that $P_c(x') = P(G(z))$ if $x' \in \mathcal{M}$ and 0 otherwise. Substituting $P_c(x')$ in Eq. 3 and after changing the order of integration we get (see Appendix A.2)

$$P(x;\sigma) = \int_{\mathbb{R}^d} P_\sigma(x|G(z))P(z)dz. \qquad (4)$$

Note that the integration now takes place in the *latent space*. In the limit $\sigma \to 0$, only the point on the manifold closest to $x$ contributes significantly, and for small enough $\sigma$ we obtain

$$P_{\sigma_d}(x|G(z)) \propto P_d(x) = \frac{1}{Q_d}\exp\left(-\frac{\mathcal{D}_\mathcal{M}^2(x)}{2\sigma_d^2}\right), \qquad (5)$$

where $Q_d$ is a normalization factor. In this limit we obtain "Energy-based model" where $E = \mathcal{D}_\mathcal{M}^2$. It is also worth mentioning that learning directly the distance $\mathcal{D}_\mathcal{M}$ to the manifold makes the algorithm time/noise condition free. This distance encapsulates the noise/time approximation using the actual quantity of interest the distance to the manifold of clean images. It therefore resolves another challenge associated with using diffusion models for image restoration (Sun et al., 2025). In this formulation of $P(x;\sigma)$, the probability at $x$ is obtained by integrating contributions from all points on the manifold, where the conditional probability depends solely on the distance to the manifold and is thus purely geometric. Each contribution is weighted by $P(z)$, which represents the likelihood that the point $G(z)$ on the manifold corresponds to a clean image. Since the distribution $P(z)$ is unknown, we estimate it using a kernel density method (Rosenblatt, 1956)(Parzen, 1962) a.k.a. ideal denoiser with delta mixture distribution / empirical distribution (Wang, 2024)(Karras et al., 2020):

$$P(z) \approx P_{\text{ker}}(z) = \frac{1}{Q_{\text{ker}}} \sum_{\alpha \in S} \exp\left(-\frac{\|z-z_\alpha\|^2}{2\sigma_{\text{ker}}^2}\right), \qquad (6)$$

where $S$ is the set of latent code indices corresponding to clean images, and $Q_{\text{ker}}$ is the normalization constant. Note that $\sigma_{\text{ker}}$ is a hyperparameter that should be chosen carefully. In Fig. 6, we illustrate $P_{\text{ker}}(z)$. Clearly, the encoding of a generic image $x$ in the latent space, i.e., $F(x)$, may lie in a region with low probability. The probability of a point $x$ being an image depends on its distance to every point on the manifold, weighted by the probability of that point in the latent space. Using this kernel approximation together with the conditional probability from Eq. (2), we can thus approximate the probability function $P(x;\sigma)$ as

$$P(x;\sigma) \approx \hat{P}(x;\sigma) = \frac{1}{Q_d Q_{\text{ker}}} \sum_{\alpha \in S} \int_{\mathbb{R}^d} \exp\left(-\frac{\|x-G(z)\|^2}{2\sigma^2}\right)\exp\left(-\frac{\|z-z_\alpha\|^2}{2\sigma_{\text{ker}}^2}\right)dz. \qquad (7)$$

## 3 GEOMETRIC VIEW OF DIFFUSION MODELS

Since the domain of both the encoder $F$ and the distance $\mathcal{D}_\mathcal{M}$ is the ambient space $\mathbb{R}^D$, effectively training mappings that enable the diffusion-like flow from corrupted images back

to clean ones on the manifold requires sampling the high-dimensional ambient space, which is an inherently challenging task due to the curse of dimensionality. Following the approach of diffusion models, we generate ambient samples by adding Gaussian noise to the data points. While this sampling strategy does not cover all possible corruptions, it empirically produces useful mappings. Notably, although the models are trained using Gaussian noise, they generalize well to other types of image corruption during testing.

To connect a corrupted image to its clean projection we use the concept of the score. The score is a $D$-dimensional vector field defined by $s(x) = \nabla_x \log P(x)$, which points in the direction of the steepest ascent of the probability density. For the distance-based probability distribution $P_d(x)$ defined in Eq. (5), we obtain:

$$s_d(x) = \nabla_x \log P_d(x) = \frac{\nabla_x P_d(x)}{P_d(x)} = -\frac{1}{\sigma_d^2} \mathcal{D}_{\mathcal{M}}(x) \nabla_x \mathcal{D}_{\mathcal{M}}(x). \tag{8}$$

Since $\mathcal{D}_{\mathcal{M}}(x)$ is the distance to the manifold, its gradient is a unit vector that points to the closest point on the manifold. Therefore, for $\sigma_d = 1$ we have:

$$x^{\text{shift}} := x + s_d(x) = x - \mathcal{D}_{\mathcal{M}}(x) \nabla_x \mathcal{D}_{\mathcal{M}}(x) = G(F(x)) = x^*, \tag{9}$$

where $x^*$ is the point on the manifold closest to $x$ (see Fig. 2). The point $x^* = x^{\text{shift}} = G(F(x))$ is known as the *ideal denoiser* in for example (Kadkhodaie et al., 2023). To incorporate the probability distribution of clean images on the manifold (or equivalently, in the latent space), we interpret the probability in the ambient space as a marginal distribution. This allows the approximation of the score function using a kernel-based method:

$$s(x) = \nabla_x \log P \approx \nabla_x \log \hat{P} =: \hat{s}(x).$$

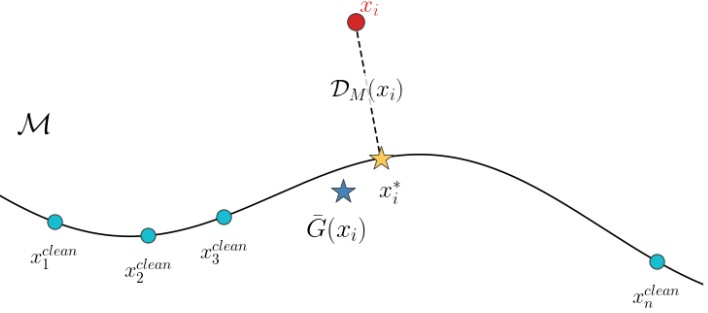

Figure 2: The manifold $\mathcal{M}$ is illustrated as the curved line. $x_i^*$ is the closest point to $x$ on the manifold. $\bar{G}(x)$ is depicted as well and is not necessarily a point on the manifold.

Direct computation results in

$$\hat{s}(x) = -\frac{1}{2\sigma_d^2} \left( x - \bar{G}(x) \right), \tag{10}$$

where $\bar{G}(x) = \sum_{\alpha \in S} \bar{G}_\alpha(x)$, and

$$\bar{G}_\alpha(x) = \frac{1}{\hat{P}(x) Q_d Q_{\text{ker}}} \int \left[ G(z) P(x|G(z)) \exp\left( -\frac{||z - z_\alpha||^2}{2\sigma_{\text{ker}}^2} \right) \right] dz. \tag{11}$$

Note that $\bar{G}(x)$, which is the (normalized) mean of $G(z)$ over the manifold (with a parameter $x$), does not necessarily lie on the manifold. In contrast, $x^* = G(F(x))$ is, by definition, a point on the manifold.

Compared to prior work, $\bar{G}$ in Eq. 10 is the same object as $\mathbb{E}_{y \in P_c}[y \mid x]$ in Kadkhodaie et al. (2023). This quantity is intractable in Kadkhodaie et al. (2023) because the integral can't be approximated by sampling clean images beyond the points in the data set. It is

therefore often replaced by the denoiser $x^*$. See Fig. 2 for an illustration and Fig. 8 for a synthetic example. Clearly, the approximation of $\bar{G}$ by $x^*$ is justified only under a uniform distribution over the manifold. In contrast, in our geometric formulation the integral over $z$ in the computation of $\bar{G}_\alpha(x)$ can be directly approximated by randomly sampling the normal distribution centered around the training point $z_\alpha$ (see details in Appendix A).

A noisy or corrupted image $x$ can be viewed as a point in the ambient space. The image generation then becomes the task of finding an appropriate, though not necessarily orthogonal, projection of this point onto the manifold of clean, meaningful images. If the mappings and functions $G$, $F$, and $\mathcal{D}_\mathcal{M}$ are perfectly accurate, a single step can move $x$ closer to the corresponding clean image. Since the ambient space is sampled sparsely, especially in regions far from the manifold, the approximations of these mappings become less accurate as the distance from the manifold increases. To address this, we employ multiple iterative steps, gradually improving accuracy as we move closer to the manifold. This process resembles a diffusion-like flow; see Fig. 8 for an illustrative example. Equations 8 and 9 motivate a diffusion-like process guided by the distance function. The score defines a vector field in the ambient space. A step in the direction of the closest point on the manifold by using the Tweedie formula (Efron, 2011) is:

$$x^{n+1} = x^n - \alpha \mathcal{D}_\mathcal{M}(x^n)\nabla_x \mathcal{D}_\mathcal{M}(x^n)/|\nabla_x \mathcal{D}_\mathcal{M}(x^n)| \quad \text{with} \quad 0 < \alpha < 1 \quad \text{and} \quad x^0 = x. \quad (12)$$

Because of the approximate nature of the distance network, we normalize the gradient in order to better control the step size. Equation 12 does not take into account the distribution of training points on the manifold. To address this limitation, we combine it with the score of the kernel method to obtain by the Tweedie formula (see Appendix A.4):

$$x^{n+1} = (1-\beta)x^n + \beta \bar{G}(x^n) - \alpha \mathcal{D}_\mathcal{M}(x^n)\nabla_x \mathcal{D}_\mathcal{M}(x^n)/|\nabla_x \mathcal{D}_\mathcal{M}(x^n)|, \quad (13)$$

where $0 < \alpha, \beta, \alpha+\beta < 1$, and $x^0 = x$. The trajectory of $x$ as it moves towards the manifold is illustrated in Fig. 8 in Appendix D.

## 4 METHODS

### 4.1 MANIFOLD-PROBABILISTIC PROJECTION MODEL (MPPM)

The autoencoder and the distance function are implemented as separate neural networks and are jointly trained using the loss function in Appendix A.1. Algorithm 1 outlines the training procedure using the clean dataset $\mathcal{X}^{\text{clean}}$ and the reconstruction of a noisy point $x$ in the ambient space. The algorithm is demonstrated for the simple case of a non-uniform distribution on the circle embedded in $\mathbb{R}^3$ in Fig. 9 in Appendix D. All the experimental and optimization details can be found in appendices C and D.

---
**Algorithm 1** MPPM

**function** TRAIN($\mathcal{X}^{\text{clean}}, \epsilon \sim \mathcal{N}(0, \sigma_d^2)$)
    $G, F, \mathcal{D}_\mathcal{M} \leftarrow \text{Train}(\mathcal{X}^{\text{clean}}, \epsilon, \mathcal{L}(F, G, \mathcal{D}_\mathcal{M}))$
**end function**
**function** RECONSTRUCTION($x, \mathcal{X}^{\text{clean}}, \alpha, \beta$,num_steps)            $\triangleright\ 0 < \alpha, \beta, \alpha+\beta < 1$
    $x^1 \leftarrow x$
    **for** $n \leftarrow 1$ to num_steps **do**
        $x^{n+1} \leftarrow (1-\beta)x^n + \beta \sum_\alpha \bar{G}_\alpha(x^n) - \alpha \mathcal{D}_\mathcal{M}(x^n)\nabla_x \mathcal{D}_\mathcal{M}(x^n)/|\nabla_x \mathcal{D}_\mathcal{M}(x^n)|$   by 13, 11
    **end for**
    **return** $x^{n+1}$
**end function**
---

### 4.2 LATENT MPPM (LMPPM)

The key difference between the pixel space and the latent space is that, in the latter, we do not assume that encoded clean and meaningful images lie on a lower-dimensional manifold. Instead, we treat the set of encoded clean and meaningful images as a point cloud that

occupies the full dimension of the latent space. We model this set as samples from a probability distribution $P(z)$. Let the set of clean and meaningful images be $\mathcal{X}^{\text{clean}}$ and the set of these encoded images be $S = \{F(\mathcal{X}^{\text{clean}})\}$. In this context, $S$ serves the role that the manifold $\mathcal{M}$ played in the previous section, in the sense that the distance function $\mathcal{D}_S$ is now computed *in the latent space* with respect to the set $S$. Let $x \in \mathbb{R}^D$ be an image and $z = F(x) \in \mathbb{R}^d$ its latent representation. The reconstructed image is then given by $\hat{x} = G(z)$. Let us define a distance function $\mathcal{D}_S : \mathbb{R}^d \to \mathbb{R}$ such that $\mathcal{D}_S(z)$ measures the distance from $z$ to the set $S$ in the latent space. Using this, we define a shift in the latent space as: $z^{\text{shift}} := z - \mathcal{D}_S(z)\nabla_z \mathcal{D}_S(z)/|\nabla_z \mathcal{D}_S(z)|$. The loss function is then given by

$$\mathcal{L}(F, G, \mathcal{D}_S) = \lambda_1 \sum_{z_i \notin S} \left(\mathcal{D}_S(z_i) - \|z_i - z_i^*\|\|\right)^2 + \lambda_2 \sum_{z_i \in S} \left(x_i^{\text{clean}} - G(z_i)\right)^2$$

$$\lambda_3 \sum_{z_i \in S} |\mathcal{D}_S(z_i)|^2 + \lambda_4 \sum_{z_i} \left(\mathcal{D}_S(z_i) - |\mathcal{D}_S(z_i)|\right)^2 \quad (14)$$

$$+ \lambda_5 \sum_{z_i \notin S} \left\|z_i^{\text{shift}} - z_i^*\right\| + \lambda_6 \sum_{z_i \notin S} \left\|G(z_i^{\text{shift}}) - x_i^*\right\|,$$

where $x_i^* = \arg\min_{\tilde{x} \in \mathcal{X}^{\text{clean}}} \|x_i - \tilde{x}\|$, and $z_i^* = F(x_i^*)$. These definitions ensure that a generic point $x$ in the ambient space, whose closest clean image in the dataset is $x^*$ is mapped to $z = F(x)$ such that its nearest neighbor in $S$ is $z^* = F(x^*)$. It is important to note that the set $S$ evolves over training iterations as the encoder $F$ and decoder $G$ are updated, and the distance function $\mathcal{D}_S$ is adjusted accordingly. The first three terms are the heart of the algorithm. The 4th element ensures positivity. The 5th and 6th terms improve consistency between all three networks. Ablation study empirically proves that these terms contribute to the performance of the method. By the kernel method, we obtain

$$\bar{z} = \frac{1}{Q} \sum_{x_j \in \mathcal{X}} F(x_j) \exp\left(-\frac{(z - F(x_j))^2}{2\sigma_{ker}^2}\right). \quad (15)$$

The complete procedure is described in Algorithm 2.

---

**Algorithm 2** LMPPM

---

   **function** TRAIN($\mathcal{X}^{\text{clean}}, \epsilon \sim \mathcal{N}(0, \sigma_d^2)$)
      $G, F, \mathcal{D}_S \leftarrow \text{Train}\left(\mathcal{X}^{\text{clean}}, \epsilon, \mathcal{L}(F, G, \mathcal{D}_S)\right)$    by 14
   **end function**
   **function** RECONSTRUCTION($x, \mathcal{X}^{\text{clean}}, \alpha, \beta,$num_steps)       $\triangleright$ $0 < \alpha, \beta, \alpha + \beta < 1$
      $z^1 \leftarrow F(x)$
      **for** $n \leftarrow 1$ to num_steps **do**
         $z^{n+1} \leftarrow (1 - \beta)z^n + \beta\bar{z}^n - \alpha\mathcal{D}_S(z^n)\nabla_z \mathcal{D}_S(z^n)/|\nabla_z \mathcal{D}_S(z^n)|$    by 15
      **end for**
      **return** $G(z^{n+1})$
   **end function**

---

## 5 EXPERIMENTS

We evaluated our MPPM method on synthetic data and our LMPPM method on real-world image datasets, where we simultaneously trained an autoencoder-like network for $F$ and $G$, and a different network for the distance function $\mathcal{D}_\mathcal{M}$ and $\mathcal{D}_S$. It is important to note that training was performed exclusively with Gaussian noise degradation, while at inference time we evaluated the models under a variety of other degradation types. We compared our results with standard denoising autoencoders (DAE) (Vincent et al., 2008) and latent diffusion models (LDM) (Rombach et al., 2022). For synthetic experiments, we evaluated on a one-dimensional manifold: a half-circle lying in the xy plane and embedded in $\mathbb{R}^3$. The points in the circle are sampled according to angular coordinates drawn from truncated normal distributions (see Fig. 8).

For real-world data, we experiment with MNIST (LeCun, 1998) and the SCUT-FBP5500

Figure 3: Top: Digit generation from pure noise, with an FID of 19.53 computed over 2000 images. Bottom: Progression of digit generation over 16 steps.

facial beauty dataset (Liang et al., 2018). To evaluate restoration performance, we apply three types of degradation to MNIST: Gaussian noise, downsampling (super-resolution), and elastic deformation, each at two severity levels. For SCUT-FBP5500, we consider four types of degradation: Gaussian noise, downsampling, random scribbles, and black patches (inpainting), also applied at two severity levels. We train our proposed methods and the comparison baselines to assess their performance across the different datasets. Detailed architecture specifications and hyperparameters are provided in appendices B and C. For synthetic data, we implement MPPM using MLP architectures. For MNIST, we employ a CNN-based autoencoder for both DAE and our LMPPM method, while for SCUT-FBP5500 we adopt a U-Net architecture with skip connections. In addition, we construct an extra set of skip connections from the latent space and combine them with the original skips through weighted summation (see Appendix B). The distance functions $\mathcal{D}_{\mathcal{M}}$ and $\mathcal{D}_S$ are implemented as MLPs with progressively decreasing layer sizes to perform dimensionality reduction. For LDM, we integrate the corresponding DAE backbone (in place of the autoencoder) with a standard diffusion model, using 2000 diffusion steps.

### 5.1 RESULTS

**MNIST Results:** For the MNIST dataset, we set the latent space dimension to 18 and the additive noise to $\epsilon = 0.4$. To calculate FID, we trained an MNIST classifier and computed an embedding distribution for each class. After reconstructing a degraded digit, we classified it and compared its embedding with the corresponding pre-computed class distribution. Table 1 reports the mean SSIM and FID metrics. Our method consistently outperforms both DAE and LDM baselines across all degradation types in terms of FID scores. Notably, DAE occasionally achieved higher SSIM values, although its visual results were inferior.

We additionally performed an ablation study to assess the significance of the distance network. Ablation[lmppm] corresponds to setting $\alpha = 0$ in the reconstruction process, while Ablation[dae] uses the DAE network instead of our $(F, G)$ network, also with $\alpha = 0$. As can be seen, when using the proposed network (trained with $\mathcal{D}$), the results improve compared to the DAE variant, but still remain below the performance of the full reconstruction setting ($\alpha > 0$).

Fig. 12 in Appendix D illustrates restoration examples for Gaussian noise, elastic deformation, and downsampling. Additional experiment included the generation of digits from a pure noise. We generated 200 images from random Gaussian noise and managed to obtain realistic digits (FID=19.5) as can be seen in Fig. 3.

**SCUT-FBP5500 Results:** Figure 4 shows restoration results on facial images with several degradation functions: excessive Gaussian noise, randomly missing pixels, random scribbles and over sharpening. The quantitative results in Table 2 support these visual observations, with our approach achieving consistently lower FID values across all degradation types. We set the latent dimension to 1024 and the additive noise to $\epsilon = 0.2$. While in some cases the DAE method achieves higher SSIM values, the visual quality of its reconstructions is noticeably inferior.

Additional results for Gaussian noise, downsampling, and over-sharpening for **SCUT-FBP5500** and **CelebA-HQ-256** are provided in Appendix D.

Table 1: Quantitative results on MNIST

| | Elastic 2.3 | | Elastic 1.8 | |
| | SSIM ↑ | FID ↓ | SSIM ↑ | FID ↓ |
|---|---|---|---|---|
| DAE | 0.66 | 69.36 | 0.59 | 134.60 |
| LDM | 0.64 | 66.52 | 0.58 | 124.05 |
| LMPPM (ours) | 0.63 | **12.61** | 0.59 | **16.38** |
| Ablation[lmppm] | 0.63 | 12.83 | 0.59 | 16.27 |
| Ablation[dae] | 0.17 | 522.39 | 0.15 | 527.25 |
| | Downsample 0.5 | | Downsample 0.35 | |
| | SSIM ↑ | FID ↓ | SSIM ↑ | FID ↓ |
| DAE | 0.79 | 31.66 | 0.54 | 133.66 |
| LDM | 0.75 | 31.61 | 0.53 | 128.80 |
| LMPPM (ours) | 0.67 | **11.27** | 0.52 | **22.65** |
| Ablation[lmppm] | 0.67 | 11.34 | 0.52 | 22.89 |
| Ablation[dae] | 0.17 | 521.14 | 0.13 | 504.08 |

Table 2: Quantitative results on SCUT-FBP5500

| | Miss pixels 0.04 | | Miss pixels 0.08 | | Miss pixels 0.1 | |
| | SSIM ↑ | FID ↓ | SSIM ↑ | FID ↓ | SSIM ↑ | FID ↓ |
|---|---|---|---|---|---|---|
| DAE | 0.917 | 33.90 | 0.798 | 49.00 | 0.745 | 47.94 |
| LDM | 0.914 | 27.35 | 0.798 | 41.47 | 0.738 | 44.41 |
| LMPPM (ours) | 0.881 | **16.20** | 0.862 | **23.92** | 0.832 | **34.13** |
| | Scribble 6 | | Scribble 13 | | Scribble 20 | |
| | SSIM ↑ | FID ↓ | SSIM ↑ | FID ↓ | SSIM ↑ | FID ↓ |
| DAE | 0.921 | 34.83 | 0.889 | 45.66 | 0.860 | 51.68 |
| LDM | 0.919 | 29.31 | 0.887 | 39.02 | 0.859 | 44.66 |
| LMPPM (ours) | 0.879 | **16.73** | 0.878 | **17.35** | 0.869 | **18.46** |
| | Sharpen 8 | | Sharpen 10 | | Sharpen 18 | |
| | SSIM ↑ | FID ↓ | SSIM ↑ | FID ↓ | SSIM ↑ | FID ↓ |
| DAE | 0.902 | 28.53 | 0.883 | 29.80 | 0.815 | 33.82 |
| LDM | 0.898 | 20.79 | 0.878 | 21.73 | 0.807 | 25.37 |
| LMPPM (ours) | 0.878 | **16.79** | 0.874 | **17.33** | 0.853 | **19.48** |

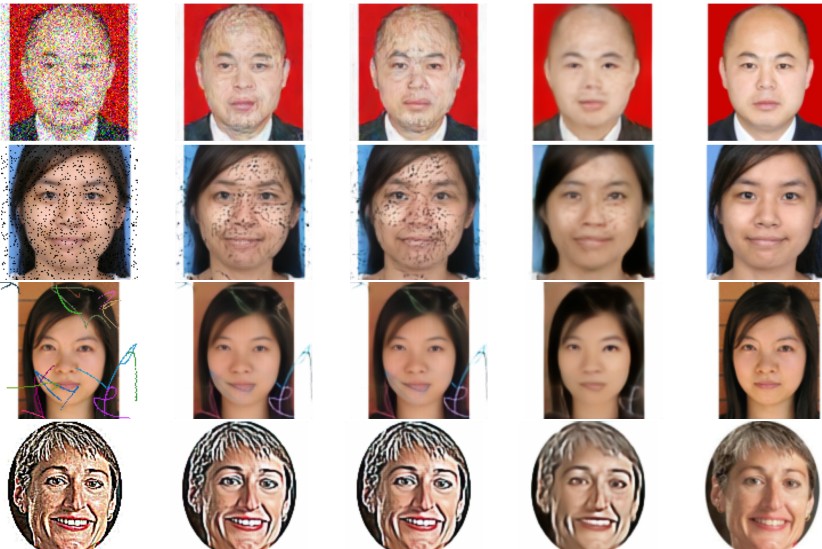

Figure 4: Variety of degradations: noise, missing pixels, scribbles and over sharpening. Left to right: degraded, DAE, LDM, LMPPM (ours), and original.



Figure 5: Left to right: over-sharpened input, DAE, LDM, LMPPM (ours), and original. LMPPM remains realistic despite changes to the face.

## 6 SUMMARY AND CONCLUSIONS

This work emphasizes the *Manifold Hypothesis* and interprets established image restoration and generation methods through a novel geometric perspective. Beyond presenting a unifying framework, which is valuable in its own right, we propose incorporating a learned distance function to the manifold. By leveraging distances to the manifold, we establish a connection between the geometric structure and a probability density approximation. By employing a kernel-like method to approximate the probability distribution on the manifold, or equivalently on the latent space, we integrate geometry and probability in a novel manner. We induce a vector field in the ambient space via the score of these probability densities. This vector field directs each point toward the manifold of clean images, considering both the structure and the distribution of clean and meaningful images on the manifold.

In this work, we utilize a (denoising) autoencoder in conjunction with the distance function. Providing an approach where both $F$ and $G$ define the manifold while maintaining their coupling to the distance function $\mathcal{D}$ from it. However, due to potential errors in the outputs of the three networks $G$, $F$ and $\mathcal{D}$, especially when $x$ is far from the manifold, this vector field is not exact. Therefore, rather than applying a single-step (weighted) projection onto the manifold, we proceed iteratively, advancing in small steps along the noisy vector field. We are currently exploring an analogous approach where VAE and GAN are coupled with the distance function. A key practical advantage of our approach is its application in the latent space. This dimensionality reduction significantly enhances the accuracy of the distance function, thereby improving restoration and generation results. Indeed, as shown in our experiments (Section 5.1), comparisons with other leading methods indicate the superior performance of our methods, particularly under severe distortions for different data sets and different distortions.

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

## A  APPENDIX: DETAILED THEORY

### A.1  MPPM

The loss function for the MPPM is

$$\mathcal{L}(F, G, \mathcal{D}_\mathcal{M}) = \lambda_1 \sum_{x_i \notin \mathcal{M}} (\mathcal{D}_\mathcal{M}(x_i) - \|x_i - x_i^*\|])^2 + \lambda_2 \sum_{x_i \in \mathcal{M}} \left( x_i^{\text{clean}} - G(F(x_i^{\text{clean}})) \right)^2$$
$$+ \lambda_3 \sum_{x_i \in \mathcal{M}} |\mathcal{D}_\mathcal{M}(x_i)|^2 + \lambda_4 \sum_{x_i \in \mathbb{R}^D} (\mathcal{D}_\mathcal{M}(x_i) - |\mathcal{D}_\mathcal{M}(x_i)|)^2 + \lambda_5 \sum_{x_i \in \mathbb{R}^D} \left( x_i^{\text{shift}} - x_i^* \right)^2,$$

$$(16)$$

where $x^* = G(F(x))$. The first term defines the distance function assuming a perfect autoencoder; the second is the standard autoencoder loss. The third term enforces the boundary condition on the distance function and the fourth ensures its positivity. The last term enforces the geometric consistency of Eq. (9) (see also Fig. 1).

### A.2  KERNEL METHOD

Detailed computation of eq. 4. A general point on the ambient space is denoted $x$ and a point of clean data is denoted $y$. The probability density of the clean images is denoted $P_c(y)$

$$P(x) = \int_\mathcal{M} P(x, y) dy = \int_{\mathbb{R}^D} P(x|y) P_c(y) dy$$
$$= \int_{\mathbb{R}^D} P(x|y) \left( \int_\mathcal{M} P(G(z)) \delta(y - G(z)) dV_\mathcal{M} \right) dy$$
$$= \int_\mathcal{M} P(x|y = G(z)) P(G(z)) \underbrace{\sqrt{g} dz}_{dV_\mathcal{M}} = \int_{\mathbb{R}^d} P(x|G(z)) P(z) dz.$$

Here $dV_\mathcal{M} = \sqrt{g} dz$ is the manifolds volume element, where $g = \det G$ and $G_{\mu\nu} = \sum_{i=1}^D J_\mu^i J_\nu^i$ is the induced metric, with the Jacobian of the embedding map given by $J_\mu^i = \partial G^i(z)/\partial z_\mu$. In the last equality, we use the identity $P(G(z)) = P(z)(\sqrt{g})^{-1}$.

### A.3 SCORE FUNCTION

$$s(x) = \nabla_x \log P \approx \nabla_x \log \hat{P} =: \hat{s}(x).$$

Direct computation results in

$$\hat{s}(x) = -\frac{1}{2\sigma_d^2} \left( x - \bar{G}(x) \right), \tag{17}$$

where $\bar{G}(x) = \sum_{\alpha \in S} \bar{G}_\alpha(x)$, and

$$\bar{G}_\alpha(x) = \frac{1}{\hat{P}(x)Q_d Q_{\text{ker}}} \int \left[ G(z)P(x|G(z)) \exp\left( -\frac{||z - z_\alpha||^2}{2\sigma_{\text{ker}}^2} \right) \right] dz. \tag{18}$$

Specifically,

$$\hat{s}(x) = \frac{1}{\hat{P}} \nabla_x \hat{P} = \frac{1}{\hat{P}} \nabla_x \left( \int P(x \mid G(z)) P_{\text{ker}}(z) dz \right).$$

Now,

$$\left( \nabla_x P(x \mid G(z)) \right) P_{\text{ker}}(z) = -\frac{1}{2\sigma_d^2 Q_d} (x - G(z)) \exp\left( -\frac{||x - G(z)||^2}{2\sigma_d^2} \right) \frac{1}{Q_{\text{ker}}} \sum_{\alpha \in S} \exp\left( -\frac{||z - z_\alpha||^2}{2\sigma_{ker}^2} \right). \tag{19}$$

The integral of $z$ in the computation of $\bar{G}_\alpha(x)$ is approximated by randomly sampling the normal distribution centered around the training point $z_\alpha$. Explicitly, we approximate the mean using and average over $n$ samples from $P_{\text{ker}}$

$$\int \left[ G(z)P(x|G(z)) \exp\left( -\frac{||z - z_\alpha||^2}{2\sigma_{\text{ker}}^2} \right) \right] dz \approx \frac{1}{n} \sum_{z_i \in \mathcal{N}(z_\alpha, \sigma_{\text{ker}}^2)} G(z_i) \exp\left( -\frac{||x - G(z_i)||^2}{2\sigma_d^2} \right), \tag{20}$$

where $\alpha$ denotes an index in the training set (see Fig. 6). The calculation of $\bar{G}_\alpha(x)$ requires evaluating $P_{\text{non-u}}(x)$ in the denominator. In particular, we approximate

$$\int \left[ P(x|G(z)) \exp\left( -\frac{||z - z_\alpha||^2}{2\sigma_{\text{ker}}^2} \right) \right] dz \approx \frac{1}{n} \sum_{z_i \in \mathcal{N}(z_\alpha, \sigma_{\text{ker}}^2)} \exp\left( -\frac{||x - G(z_i)||^2}{2\sigma_d^2} \right). \tag{21}$$

Note that in the computation of $\bar{G}_\alpha$ all constant factors $Q_d$, $Q_{\text{ker}}$ and $\frac{1}{n}$, are canceled between the numerator and the denominator.

### A.4 THE TWEEDIE FORMULA

The "flow" equations 12 and 13 are the Tweedie formulas for the corresponding probability functions. Eq. 12 follows

$$P_d(x) = \frac{1}{Q_d} \exp\left( -\alpha \mathcal{D}_\mathcal{M}^2(x) \right), \tag{22}$$

and Eq. 13 follows

$$P(x) = \frac{1}{Z} P_d(x) \sum_{\alpha \in S} \int_{\mathbb{R}^d} \exp\left( -\beta ||x - G(z)||^2 \right) \exp\left( -\frac{||z - z_\alpha||^2}{2\sigma_{\text{ker}}^2} \right) dz. \tag{23}$$

where $Z$ is a normalization factor.

One can easily verify from Eq. 12 that we decrease the distance to the manifold along the flow. Indeed

$$\mathcal{D}_\mathcal{M}(x^{n+1}) = \mathcal{D}_\mathcal{M}(x^n - \epsilon \mathcal{D}_\mathcal{M}(x^n) \nabla \mathcal{D}_\mathcal{M}(x^n))$$
$$= \mathcal{D}_\mathcal{M}(x^n) - \epsilon \nabla \mathcal{D}_\mathcal{M}(x^n) \cdot \mathcal{D}_\mathcal{M}(x^n) \nabla \mathcal{D}_\mathcal{M}(x^n) + O(\epsilon^2) = (1 - \epsilon)\mathcal{D}_\mathcal{M}(x^n) + O(\epsilon^2)$$

where we used the fact that the distance function is a solution of the Eikonal equation $||\nabla \mathcal{D}_\mathcal{M}(x)||^2 = 1$.

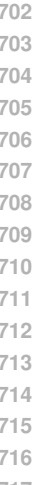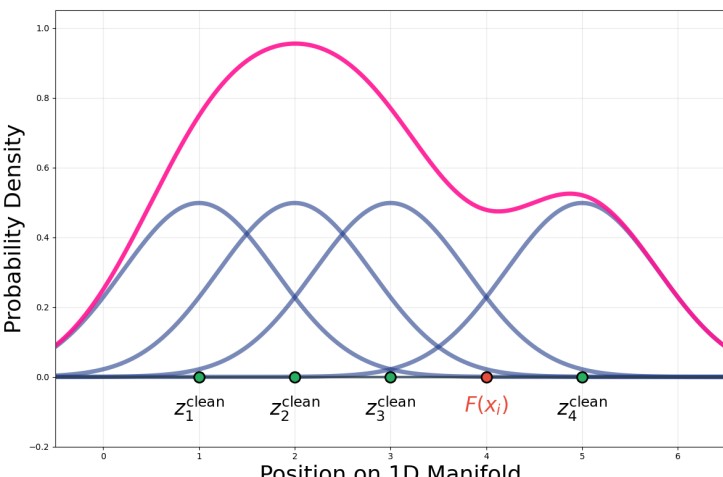

Figure 6: An illustration of the kernel approximation $P_{\mathrm{ker}}(z)$ of the probability distribution $P(z)$ in the latent space.

## B   Appendix: Detailed Experimental Setup

### B.1   Notation and Abbreviations

Here are some notations and definitions:

Table 3: Glossary of abbreviations and terms used throughout the paper

| Term | Definition |
| --- | --- |
| DAE | Denoising Autoencoder |
| MPPM | Manifold Probabilistic Projection Model (our proposed approach) |
| LMPPM | Latent Manifold Probabilistic Projection Model (our proposed approach) |
| LDM | Latent Diffusion Model |
| SSIM | Structural Similarity Index Measure |
| BN | Batch Normalization |

Table 4: Summary of experimental datasets used for evaluating restoration performance

| Dataset | Description |
| --- | --- |
| MNIST | 60,000 training/10,000 test grayscale images ($28 \times 28$ pixels) |
| SCUT-FBP5500 | 5,500 facial images with beauty scores (resized to $120 \times 120$) |

### B.2   Degradations

**Degradation Parameters**   We apply six degradation types to simulate real-world image corruption scenarios. Each degradation is applied at three severity levels (mild, intermediate, and severe) to test the robustness of restoration methods:

**Degradation Methods**   Brief descriptions of each degradation type:

- **Gaussian noise**: Additive zero-mean Gaussian noise that simulates sensor noise or transmission errors.
- **Elastic deformation**: Non-rigid distortions implemented using `torchvision.transform.ElasticTransform`($\alpha = 34, \sigma$) that simulate warping effects.

Table 5: Degradation parameters at different severity levels. Note: Lower $\sigma$ values for elastic deformation indicate more severe distortion due to increased localized displacement

| Degradation | Mild | Severe |
|---|---|---|
| Gaussian noise ($\sigma$) | 0.2 | 0.3 |
| Elastic deformation ($\sigma$) | 1.5 | 1.1 |
| Super-resolution (downsampling factor) | 0.5 | 0.35 |
| Missing Pixels | 0.04 | 0.1 |
| Number of Scribbles | 13 | 20 |
| Over Sharpening | 10 | 18 |

- **Super-resolution**: Downsampling followed by upsampling to original resolution, simulating reconstruction from low-resolution data.
- **Missing Pixels**: Set black patches with some coverage portion;
- **Scribbles** Add $n$ random scribbles with random colors
- **Over Sharpening by factor** $s$: $I = I + s(I - I * \sigma_s)$

### B.3 MODEL ARCHITECTURES

We implemented three main architectures across all experiments, with design choices tailored to each dataset's complexity.

**Synthetic Data Model** Synthetic data for MPPM experiments use MLP-based networks with a latent dimension of 8, selected based on the low intrinsic dimensionality of these manifolds:

Table 6: Network architectures for synthetic data experiments. All models use fully-connected layers

| Component | Architecture |
|---|---|
| Encoder | $3 \to 64 \to 32 \to 16 \to 8$ with ReLU |
| Decoder | $8 \to 16 \to 32 \to 64 \to 3$ with ReLU |
| Distance Network | $8 \to 64 \to 32 \to 16 \to 1$ with ReLU, dropout=0.2 |

**MNIST Models** MNIST experiments use CNN-based models with latent dimension 18, chosen to capture the variability among handwritten digits while promoting compact representations.

Table 7: Network architectures for MNIST experiments

| Component | Architecture |
|---|---|
| Encoder | $\text{Conv2d}(1 \to 32 \to 64, \text{kernel} = 3, \text{stride} = 2) \to \text{Flatten} \to \text{Linear}(64 \times 7 \times 7 \to 18)$ |
| Decoder | $\text{Linear}(18 \to 64 \times 7 \times 7) \to \text{Reshape} \to \text{ConvTranspose2d}(64 \to 32 \to 1) \to \text{Sigmoid}$ |
| Distance Network | $18 \to 100 \to 50 \to 20 \to 1$ with ReLU, dropout=0.2 |

**SCUT-FBP5500 Models** Facial image experiments employ a U-Net with skip connections and a latent dimension of 1024, which accommodates the higher complexity of facial features while enabling detailed reconstruction. Note that in the U-Net architecture, during the inference process we use iterations (denoted by superscripts) such that

$$F(x^n) = \left(S_1^n, S_2^n, \ldots, S_k^n, z^n\right)^T,$$

and

$$x^{n+1} = G\left(S_1^n + \hat{S}_1^n(z^{n+1}), \ S_2^n + \hat{S}_2^n(z^{n+1}), \ \ldots, \ S_k^n + \hat{S}_k^n(z^{n+1}), \ z^{n+1}\right).$$

Here, each $\hat{S}_i^n(z^{n+1})$ denotes the projection of the latent space $z^{n+1}$ onto the corresponding skip connection $S_i^n$. Thus, the updated skip connection is formed by adding the original

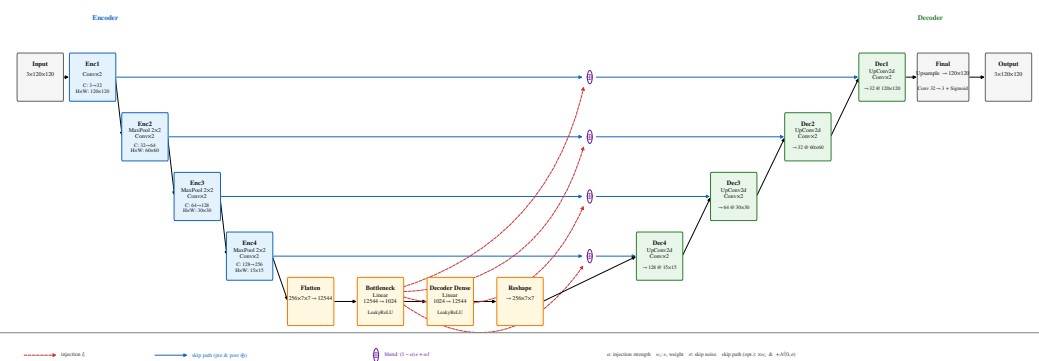

Figure 7: Modified U-net architecture

skip feature $S_i^n$ with the new projected feature $\hat{S}_i^n(z^{n+1})$ before being passed to $G$. The architecture is illustrated in Fig. 7.

| Component | Structure |
|---|---|
| EncoderBlock($C_{in} \rightarrow C_{out}$) | Conv($C_{in} \rightarrow C_{out}$) $\rightarrow$ BN $\rightarrow$ LReLU $\rightarrow$ |
| | Conv($C_{out} \rightarrow C_{out}$) $\rightarrow$ BN $\rightarrow$ LReLU $\rightarrow$ MPool |
| Encoder | EncoderBlock($3 \rightarrow 32$), output: $60 \times 60$ |
| | EncoderBlock($32 \rightarrow 64$), output: $30 \times 30$ |
| | EncoderBlock($64 \rightarrow 128$), output: $15 \times 15$ |
| | EncoderBlock($128 \rightarrow 256$), output: $7 \times 7$ |
| | Flatten $\rightarrow$ Linear($12544 \rightarrow 1024$) $\rightarrow$ LReLU |
| DecoderBlock($C_{in}, C_{skip}, C_{out}$) | ConvT($C_{in} \rightarrow C_{in}$) $\rightarrow$ Cat($[C_{in}, C_{skip}]$) $\rightarrow$ |
| | Conv($C_{in} + C_{skip} \rightarrow C_{in}$) $\rightarrow$ BN $\rightarrow$ LReLU $\rightarrow$ |
| | Conv($C_{in} \rightarrow C_{out}$) $\rightarrow$ BN $\rightarrow$ LReLU |
| Decoder | Linear($1024 \rightarrow 12544$) $\rightarrow$ Reshape($256, 7, 7$) |
| | DecoderBlock($256, 256, 128$), output: $15 \times 15$ |
| | DecoderBlock($128, 128, 64$), output: $30 \times 30$ |
| | DecoderBlock($64, 64, 32$), output: $60 \times 60$ |
| | DecoderBlock($32, 32, 32$), output: $120 \times 120$ |
| | Conv($32 \rightarrow 3$) $\rightarrow$ Sigmoid |
| Distance Network | $1024 \rightarrow 100 \rightarrow 50 \rightarrow 20 \rightarrow 1$ with ReLU, dropout=0.2 |

Table 8: Network architectures for SCUT-FBP5500 experiments. Skip connections connect corresponding Encoder and Decoder layers through concatenation. The encoder and decoder blocks are represented as parameterized functions (shown in italic font), where $C_{in}$, $C_{out}$, and $C_{skip}$ represent the number of input, output, and skip connection channels respectively. Abbreviations: Conv = Conv2d (kernel=3, padding=1), BN = BatchNorm2d, LReLU = LeakyReLU(0.2), MPool = MaxPool2d(2), ConvT = ConvTranspose2d(kernel=2, stride=2), Cat = Concatenation. The bottleneck dimension is 1024.

## C   TRAINING AND EVALUATION

Table 9: MPPM training and inference parameters for synthetic data

| Parameter | Value |
|---|---|
| Optimizer | Adam ($\beta_1 = 0.9$, $\beta_2 = 0.999$) |
| Learning rates | AE: $1 \times 10^{-3}$, Distance network: $1 \times 10^{-3}$ |
| Weight decay | $1 \times 10^{-4}$ |
| Batch size | 550 |
| Training epochs | 500 |
| Loss function | Composite loss (Equation 16) |
| Early stopping | Patience: 100 epochs |
| $\alpha$ (distance gradient step) | 0.15 |
| $\beta$ (kernel averaging weight) | 0.1 |
| Convergence tolerance | 0.005 |
| Maximum iterations | 60 |

Table 10: LMPPM training parameters across all experiments, determined through preliminary grid search, diffusion steps are define the number of steps in algorithm 2

| Parameter | Value |
|---|---|
| Optimizer | Adam ($\beta_1 = 0.9$, $\beta_2 = 0.999$) |
| Learning rates | AE: $1 \times 10^{-3}$, Distance network: $1 \times 10^{-5}$, LDM: $1 \times 10^{-3}$ |
| Batch size | MNIST: 128, SCUT-FBP5500: 32 |
| Training epochs | MNIST: 100, SCUT-FBP5500: 75 |
| Loss functions | DAE: L2, LDM: MSE, LMPPM: Composite loss 14 |
| Early stopping | Patience: 8 epochs |
| Diffusion steps | MNIST: 2000, SCUT-FBP5500: 2000 |

# D ADDITIONAL RESULTS

Here, we present additional experimental results. Figures 8 and 9 depict the results of the MPPM algorithm.

Figures 10 and 11 illustrate the advantages of the proposed MPPM method compared to the diffusion model. We used 1000 diffusion steps during both training and inference. Because the data points are not uniformly distributed, most of the diffusion models reconstructed samples concentrate in the dense region (the upper half-circle), as shown in the left panel of Figure 11. In contrast, the proposed MPPM method effectively handles this non-uniformity through our formulation, resulting in a significantly smaller reconstruction error. Pseudocode for DDPM (Denoising Diffusion Probabilistic Model) training and inference is summarized in Algorithms 3 and 4. Note the difference from Chen et al. (2024), who analyzed the trajectory of the flow by measuring the deviation from the line between the degraded image and the point found on the manifold.

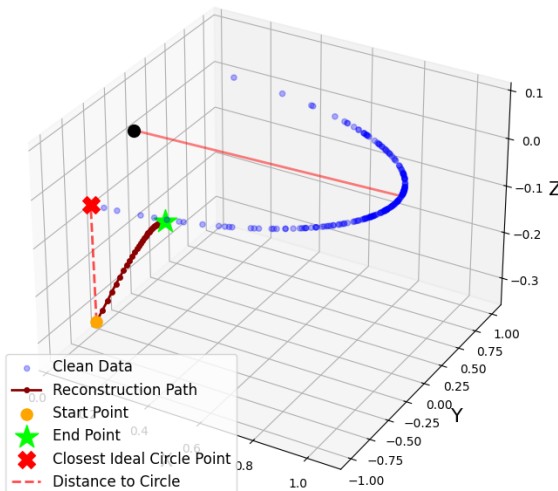

Figure 8: The manifold $\mathcal{M}$ is the unit circle lying in the xy-plane and is parametrized by the azimuth angle $\theta$. It is sampled according to a normal distribution centered at $\theta_0$ indicated by the red line. The reconstruction trajectory is shown in dark red. Note that the final result of the iterations on $x$ does not converge to $x^*$ which is the closest point on the circle. Instead, it is influenced by the data distribution on the manifold through the effect of $\bar{G}(x)$.

---

**Algorithm 3** DDPM Training

---

1: **Precompute noise schedule:**
2: $\beta_t$ (linear schedule from 0.0001 to 0.02)
3: $\alpha_t = 1 - \beta_t$
4: $\bar{\alpha}_t = \prod_{i=1}^{t} \alpha_i$
5: **for each batch do**
6:     Sample timestep $t \sim \text{Uniform}(0, T-1)$
7:     Sample noise $\epsilon \sim \mathcal{N}(0, I)$
8:     **Forward diffusion:**
$$x_t = \sqrt{\bar{\alpha}_t}\, x_0 + \sqrt{1 - \bar{\alpha}_t}\, \epsilon$$
9:     Predict noise: $\epsilon_{\text{pred}} = \text{model}(x_t, t/T)$
10:     Compute loss:
$$L = \|\epsilon - \epsilon_{\text{pred}}\|^2$$
11:     Backpropagate and update parameters
12: **end for**

---

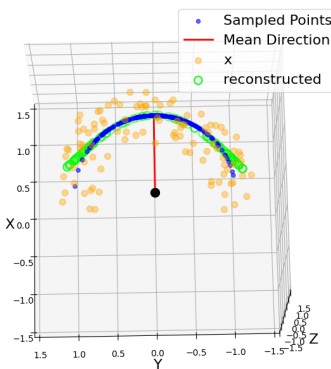 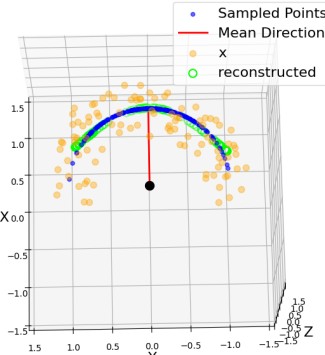

(a) DAE restoration. MSE = 0.032, max error = 0.147.

(b) MPPM restoration. MSE = 0.026, max error = 0.060.

Figure 9: Comparison between the DAE and our proposed MPPM, this example uses the same setup as in Fig. 8. The error was computed as the deviation from the unit circle in 2D. In regions of the circle with lower probability density, the DAE is more prone to error than the proposed MPPM method.

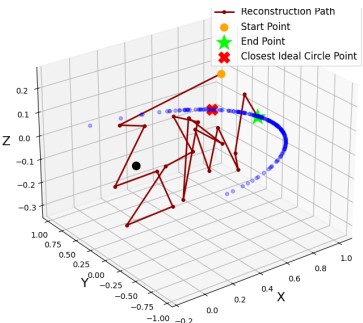 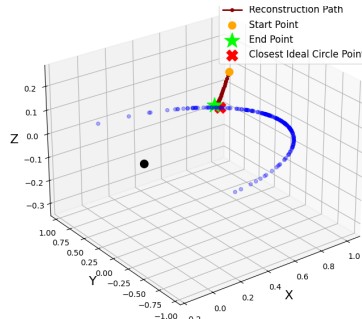

Figure 10: Left: Diffusion model trajectory, Right: MPPM trajectory

---

**Algorithm 4** DDPM Inference (Stochastic, 1000 steps)

---

1: Initialize $x_T \sim \mathcal{N}(0, I)$
2: **for** $t = T - 1$ **down to** $0$ **do**
3:     Predict noise: $\epsilon_{\text{pred}} = \text{model}(x_t, t/T)$
4:     **Denoise:**

$$x_t = \frac{x_t - \text{coef}_t \, \epsilon_{\text{pred}}}{\sqrt{\alpha_t}}$$

5:     **if** $t > 0$ **then**
6:         Sample $z \sim \mathcal{N}(0, I)$
7:         Add noise: $x_t = x_t + \sigma_t z$
8:     **end if**
9: **end for**
10: **return** $x_0$

---

Next, we present additional results on the MNIST dataset using the LMPPM algorithm. Figure 12 shows reconstruction results under noise, elastic, and downsampling deformations, compared with the DAE and LDM models. Figures 13, 14, 15, and 16 present reconstruction

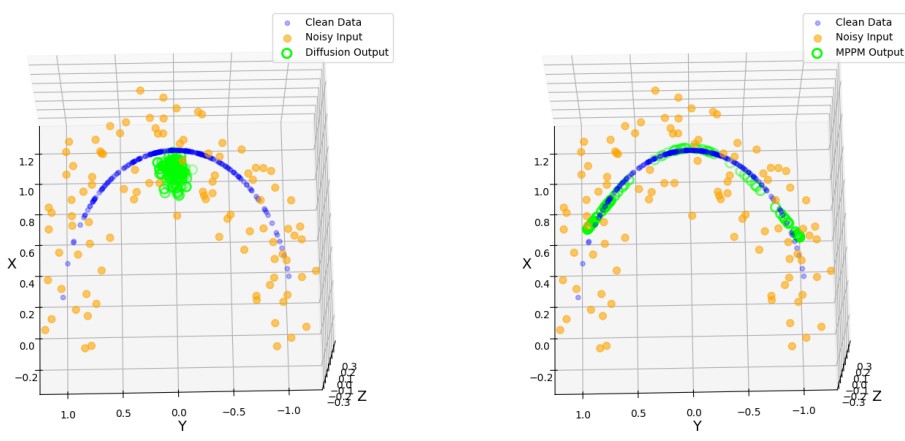

Figure 11: Left: Diffusion restoration, MSE=0.138, Right: MPPM restoration, MSE=0.026

results for missing pixels, scribbles, noise, and over sharpening deformations, respectively for the SCUT-FBP5500 dataset. We compare our method with the DAE and LDM models. Finally, Figure 17 shows the reconstruction after 4 iterations.

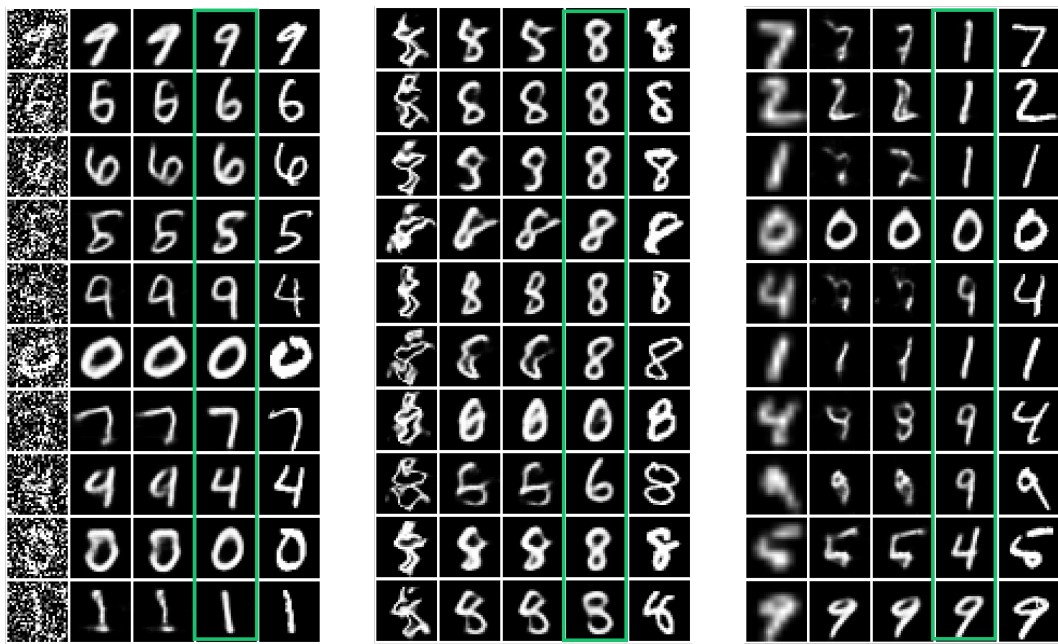

Figure 12: Left panel: noise = 0.7; middle panel: elastic ($\alpha = 0.34$, $\sigma = 1.8$); right panel: downsampling factor = 0.35. In all panels, from left to right: degraded, DAE, LDA, LMPPM (ours), and original.

### D.1    CELEBA DATASET

We applied our method to the CelebA-HQ-256 dataset. We used the same architecture of SCUT-FBP5500 model. The results are shown in Figures 18 and 19, as well as in Table 11.

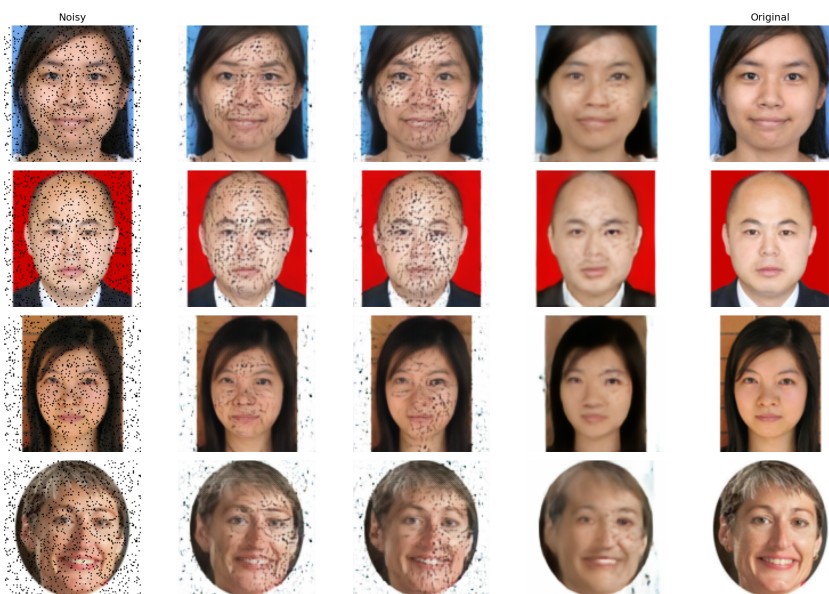

Figure 13: Missing pixels. From left to right: degraded, DAE, Diffusion, LMPPM, original.

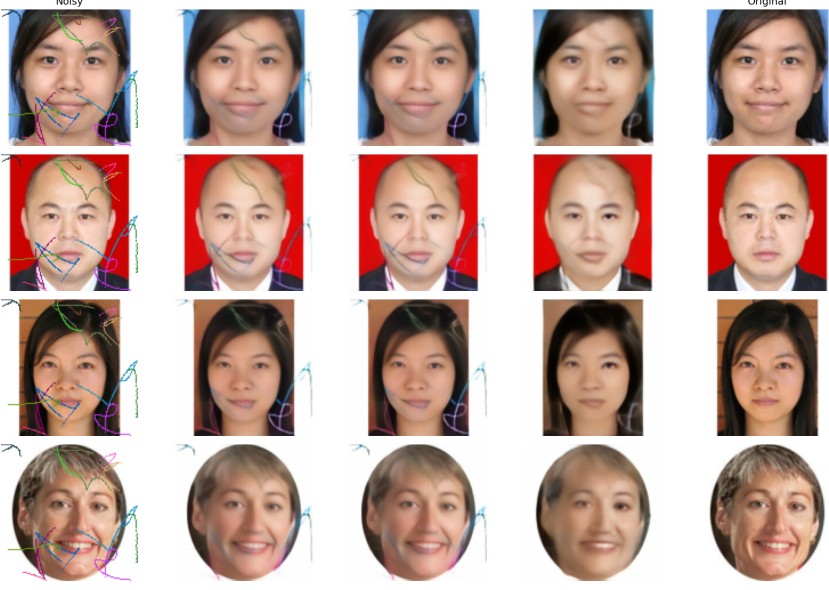

Figure 14: Images with 13 scribbles. From left to right: degraded, DAE, Diffusion, LMPPM, original.

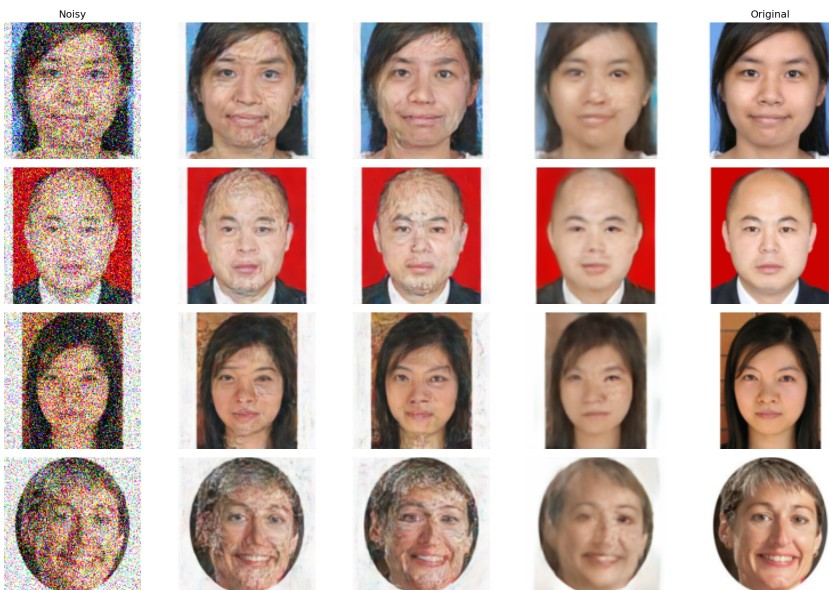

Figure 15: Noise $\sigma = 0.3$. From left to right: degraded, DAE, Diffusion, LMPPM, original.

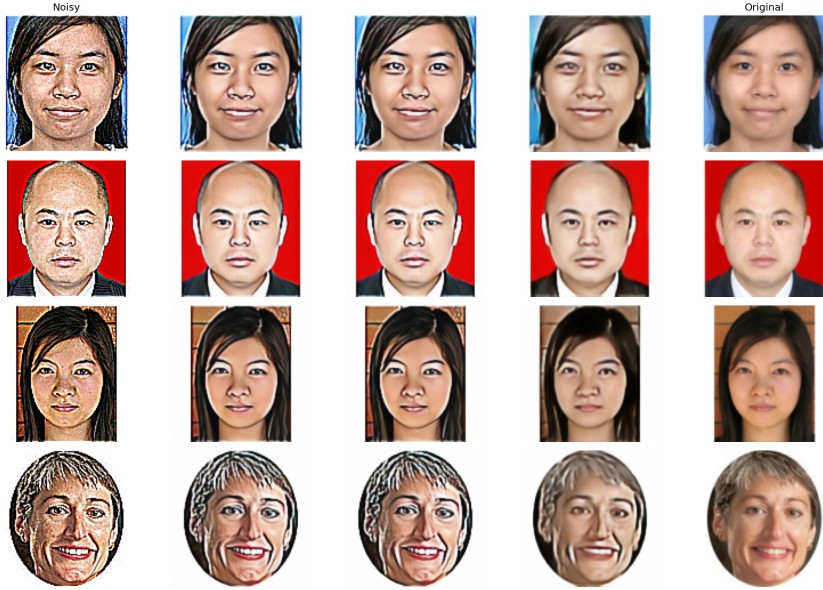

Figure 16: Over sharpening. From left to right: degraded, DAE, LDM, LMPPM, original.



Figure 17: Gradual reconstruction of missing pixels degradation.

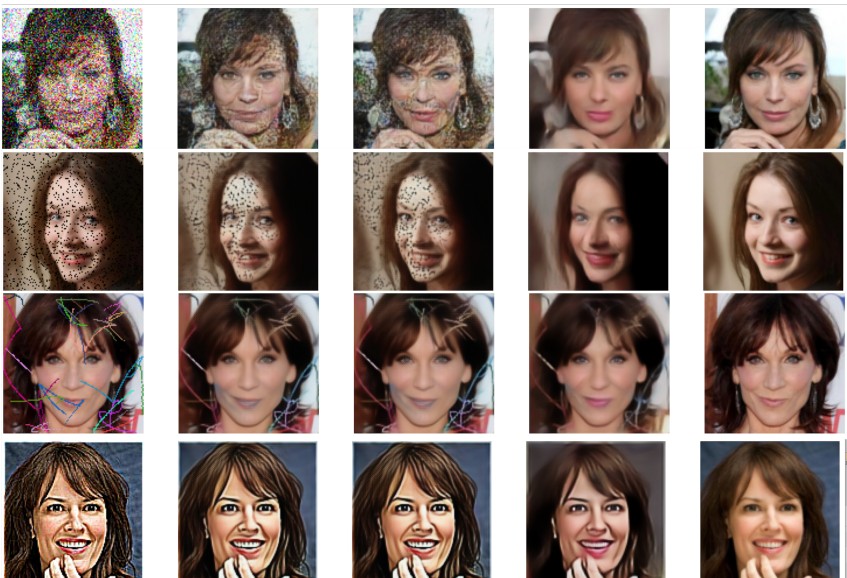

Figure 18: Different degradations applied to the CelebA-HQ-256 dataset. From left to right: degraded, DAE, LDM, LMPPM (ours), and original.

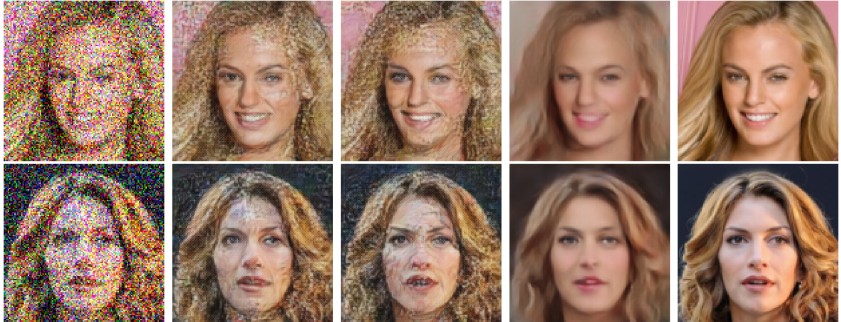

Figure 19: Excessive Gaussian noise ($\sigma = 0.3$) applied on CelebA-HQ-256 dataset. From left to right: degraded, DAE, LDM, LMPPM and original.

As evident from the quantitative results, our method achieves a significantly lower FID score, while the SSIM values remain approximately similar across all methods.

We further compared our method to DiffBIR Lin et al. (2024). DiffBIR tackles blind image restoration using two stages: (1) degradation removal, and (2) information regeneration. The first stage removes degradations and produces a high-fidelity but often over-smoothed intermediate result, while the second stage regenerates realistic textures and details. For completeness, we conducted three experiments: (i) DiffBIR after its first stage only, (ii) full DiffBIR, and (iii) our LMPPM followed by DiffBIRs second stage. The results are shown in Figure 20 and in the bottom panel of Table 11.

As can be seen, LMPPM outperforms the first stage of DiffBIR both visually (second and third columns from the left) and quantitatively, especially under the missing-pixel and scribble degradations. The output of DiffBIRs second stage is realistic and perceptually high-quality. Notably, the full DiffBIR model (fourth column from the left) performs well in removing Gaussian noise (first row), even though the reconstructed image differs from the original image (right column). The best performance is achieved by applying our LMPPM followed by DiffBIRs second stage (second column from the right), indicating that our blind degradation-removal module provides a strong foundation for high-quality restoration.

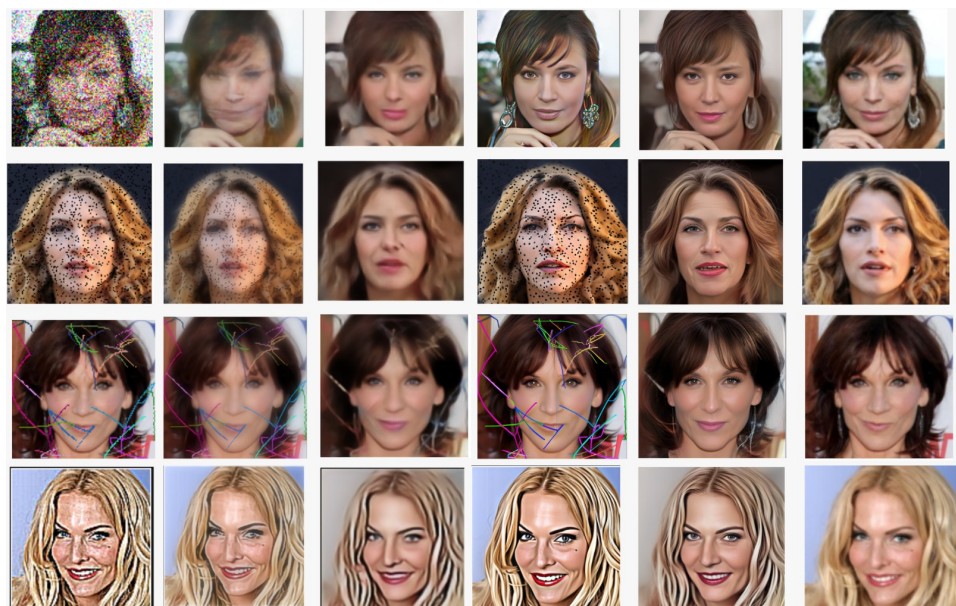

Figure 20: Comparison to DiffBIR method Lin et al. (2024). From left to right: degraded, DiffBIR stage1, LMPPM, DiffBIR (stage1 + stage2), LMPPM+DiffBIR stage2, original

Table 11: Quantitative results on the CelebA-HQ-256 dataset, compared also to the DiffBIR method Lin et al. (2024).

| | Noise 0.3 | | Scribbles 22 | | Miss Pixels 0.1 | | Sharpen 12 | |
|---|---|---|---|---|---|---|---|---|
| | SSIM ↑ | FID ↓ | SSIM ↑ | FID ↓ | SSIM ↑ | FID ↓ | SSIM ↑ | FID ↓ |
| DAE | 0.694 | 43.05 | 0.817 | 54.75 | 0.762 | 49.23 | 0.719 | 46.64 |
| LDM | 0.663 | 34.99 | 0.793 | 42.73 | 0.757 | 41.38 | 0.724 | 34.54 |
| LMPPM (ours) | 0.707 | **23.92** | 0.757 | **30.69** | 0.671 | **25.63** | 0.714 | **28.25** |
| DiffBIR | 0.70 | 24.09 | 0.69 | 42.95 | 0.58 | 42.52 | 0.76 | 31.95 |
| DiffBIR stage1 | 0.67 | 28.52 | 0.71 | 43.85 | 0.69 | 44.01 | 0.89 | 28.55 |
| LMPPM+DiffBIR stage2 | 0.68 | **22.64** | 0.69 | **30.68** | 0.63 | **23.21** | 0.72 | **25.69** |

## E    VALIDATION OF FID METRIC IMPLEMENTATION

Given the challenging nature of the degradation tasks presented in this paper, baseline methods such as LDM and DAE yielded relatively high FID scores. To ensure these values reflect true performance rather than an artifact of the metric implementation, we conducted a validation experiment.

We utilized the MNIST dataset with minimal deformations to test the sensitivity of our FID calculation. As shown in the table below, our evaluation pipeline correctly reports low FID scores in this simplified regime. This confirms the reliability of our metric and suggests that the performance gaps observed in the main experiments are driven by model capabilities on complex data, rather than measurement errors.

Table 12: FID values for different small degradations of MNIST

| | Low Severity ($\sigma$) | DAE | LDM | LMPPM |
|---|---|---|---|---|
| Elastic | 5.5 | 7.11 | 8.36 | 9.57 |
| Noise | 0.1 | 3.12 | 3.65 | 9.99 |
| Down sample | 0.9 | 5.06 | 5.84 | 9.78 |

## F  Use of Large Language Models

Large Language Models (LLMs) were used in this work solely as a language assistance tool for English polishing and proofreading. Specifically, we employed LLMs to:

- Improve grammar, syntax, and sentence structure in the manuscript
- Enhance clarity and readability of technical descriptions
- Correct spelling and typographical errors
- Suggest more precise word choices and phrasing

The LLMs did not contribute to research ideation, methodology development, experimental design, data analysis, or the generation of scientific content. All research concepts, approaches, results, and conclusions presented in this paper are entirely the work of the human authors. The LLMs were used exclusively for language refinement of content that was already conceptualized and written by the authors.

Additionally, an LLM was used to assist in drafting this disclosure section itself, based on the authors' description of how LLMs were employed in the research process.

We take full responsibility for all content in this manuscript, including any text that was refined with LLM assistance. All factual claims, scientific interpretations, and conclusions remain our own work and responsibility.

