# OpenReview forum: "A Geometric Unification of Generative AI with Manifold-Probabilistic Projection Models"
_ICLR.cc/2026/Conference — Submitted to ICLR 2026_

### Official Review · Reviewer_Vt93 · 2025-10-15

**Soundness:** 3
**Presentation:** 2
**Contribution:** 2
**Rating:** 4
**Confidence:** 4

**Summary:**

The article presents an integrated perspective that unifies geometric and probabilistic views by introducing a geometric framework and a kernel-based probabilistic method. Within this framework, the diffusion model is interpreted as a projection mechanism on the manifold of “high-quality images,” providing new insight into its underlying nature. Building on this interpretation, the authors propose a deterministic model—the Manifold Probability Projection Model (MPPM)—which operates coherently in both the representation (pixel) and latent spaces. Experimental results indicate that the Latent Space MPPM (LMPPM) surpasses the latent diffusion model (LDM) across multiple datasets, demonstrating superior performance in image restoration task.

**Strengths:**

1.The perspective of this article is very interesting. It is of great significance to unify the understanding of geometry and probability. This is of great significance for modeling more complex data manifold distributions.

2.The theory in this article is very solid. As a work of great theoretical significance, it deserves attention.

3.The paper is well-written and the motivation is very convincing.

**Weaknesses:**

1.I have some concerns about the theoretical assumptions.  The article assumes the existence of Gaussian noise perturbations between points on the clean image manifold and the real images.  However, if the task is not image restoration, or if the data are already sufficiently clean, this assumption and consequently the proposed theory may not hold effectively.

2.The experimental section lacks comparisons with several relevant baselines in image restoration and inverse problem research [1–3]. In addition, manifold-preserving approaches [4–6] should also be considered for a more comprehensive evaluation. It seems insufficient that the author only compares with DAE and LDM.

[1] A Unified Conditional Framework for Diffusion-based Image Restoration

[2] DiffBIR: Towards Blind Image Restoration with Generative Diffusion Prior

[3] Refusion: Enabling Large-Size Realistic Image Restoration

[4] Manifold Preserving Guided Diffusion

[5] CFG++: Manifold-Constrained Classifier-Free Guidance for Diffusion Models

[6] Improving Diffusion Models for Inverse Problems using Manifold Constraints

3.Why do other methods work well on SSIM but worse on FID? Could this be because they only learned the distribution with noise added instead of the clean data distribution? The author lacks a more profound analysis.

**Questions:**

1.Would the proposed method still be effective under the assumption of clean data? Or whether it can be directly used for generating images rather than for image restoration tasks?

2.Refer to Weakness 2, how about the performance of other related models?

3.In Figure 5, I noticed that the image generated by LMPPM does not contain white teeth. Could this be caused by the limitations of some manifold probability distributions? Or is it because of some probability assumptions that the model ignores the special manifold of the tooth part?

---

> ### Author Response · Authors · 2025-11-21
> **Response to Reviewer Vt93**
>
> We thank the reviewer for the positive evaluation of our theory and motivation and for the interesting remarks and intriguing questions.
>
> 1. **Assumptions (Gaussian Noise & Clean Data)**
>
> The reviewer asked about the assumption of Gaussian noise and the necessity of clean data. Our method assumes the existence of a manifold of clean images (defined by the AE trained on clean data). However, points outside the manifold are treated as degraded images. Importantly, we do not require the degradation to be Gaussian. The probability function we construct (Eq. 4-7) creates a vector field in the ambient space that directs any point towards the manifold. This is why our method succeeds on "Scribbles" and "Missing Pixels" (Fig. 4) despite never seeing these corruptions during training. The score directs the flow to the manifold regardless of the corruption type.
>
> 2. **Comparison Baselines (DiffBIR, etc.)**
>
> A full comparison to all of them is challenging due to the scope, but the key distinction is that our method achieves blind restoration in a self-supervised manner, whereas many baselines (like DiffBIR) rely on large-scale priors or paired training. If time permits we will compare to one of these baselines.
>
> 3. **FID vs. SSIM Discrepancy**
>
> The reviewer asked why some methods achieve better FID but worse SSIM. SSIM can sometimes be misleading because it measures local similarity between the clean image and the restored one. In cases involving color shifts, over-smoothing, or small variations in facial features, the SSIM score may decrease, even when the reconstruction is perceptually valid. Additionally, the $\bar{G}$ formulation may guide the restoration toward a slightly different but still plausible image (see Fig. 5 for example). Nevertheless, both visually and in terms of FID, our method outperforms the alternatives.
>
> 4. **Artifacts (Teeth Color)**
>
> Response: Regarding the "pink teeth" artifact in Fig.5: There is no probabilistic assumption that ignores teeth. The result is simply a consequence of the algorithmic process. Our interpretation is that (a) the projection step adjusts the image toward the learned manifold, and in this particular case it caused the teeth and lip colors to blend slightly, producing a pinkish tone, and (b) the $\bar{G}$ function can introduce small shifts in facial attributes, such as a tendency toward non-smiling expressions.

---

> > ### Comment · Reviewer_Vt93 · 2025-11-26
> > **Reply to Rebuttal**
> >
> > Thanks for the authors' reply. I still have the concern about Weakness 1. Intuitively, the proposed method does not rely on Gaussian degradation and may be generalized to more cases. However, as a highly theoretical paper, the author should at least consider providing theoretical proof and analysis for a broader range of situations. For example, attempting to extend the theory to the probability distribution family?

---

> ### Author Response · Authors · 2025-11-28
>
> The probability of a non data point $x$ is designed such that the resulted score vector field points towards the manifold and its more densely populated region. We therefore choose naturally the conditional probability of the corrupted image $x$ conditioned on clean data point $x'$ as $P(x|x') =f(D(x,x'))$ where $D(x,x')$ is the distance between the clean and corrupted image. $f$ is a monotonically decreasing function. For ease of analysis and computation we choose $f$ to be a Gaussian. The second assumption is the usual one $D(x,x')=||x-x'||$. These considerations lead to the following expression for the conditional probability function
> \begin{equation}\label{cond-prob}
> P_{\sigma}({x}| x') = \frac{1}{Q_d}\exp\left({-\frac{\|{x}-x'\|^2}{2\sigma^2}}\right).
> \end{equation}
> and the probability on the ambient space is then  the well-known expression \citep{Kadkhodaie-etal2023}\citep{Sun-etal2025}
> \begin{equation}
> \label{eq:prob-ambient}
> P({x};\sigma)=\int_{\mathbb{R^D}} P_\sigma({x}| x')P_c(x')dx'=\frac{1}{Q_d}\int_{\mathbb{R^D}}\exp\left({-\frac{\|{x}-x'\|^2}{2\sigma^2}}\right)P_c(x')dx'.
> \end{equation}
> Note that even though the Gaussian makes it look like we deal only with Gaussian noise it actually has no assumption on the kind of degradation that created $x$ from $x'$. It only assumes that the probability of this event decreases exponentially with the distance between the points. The result is a blind image denoising that does not need to have the type of noise or its amplitude as input.
>
> This clarification was added in the paper in Section 2.
>
> We additionally compared our method to DiffBIR, ECCV 2024 paper, which addresses blind image restoration using two stages: (1) degradation removal and (2) information regeneration. The first stage removes degradations and yields a high-fidelity but often over-smoothed result, while the second stage synthesizes realistic textures and details.
> To provide a fair comparison, we evaluated: (i) DiffBIR after the first stage only, (ii) full DiffBIR, and (iii) our LMPPM followed by DiffBIR’s second stage. Results appear in Fig.20 and the bottom panel of Table 11 in Appendix D1..
> As shown, LMPPM outperforms DiffBIR’s first stage both visually (second and third columns) and quantitatively, particularly under missing-pixel and scribble degradations. DiffBIR’s second stage produces realistic textures, but its full output (fourth column) can deviate from the ground truth (right column), even when it removes Gaussian noise effectively (first row).
> The best overall performance is obtained by applying our LMPPM followed by DiffBIR’s second stage (second column from the right), indicating that our degradation-removal module provides a stronger and more reliable initial reconstruction.

---

### Official Review · Reviewer_UwHW · 2025-10-27

**Soundness:** 2
**Presentation:** 2
**Contribution:** 2
**Rating:** 4
**Confidence:** 2

**Summary:**

The paper proposes a new model, the Manifold Probabilistic Projection Model (MPPM) and its latent version, which interprets diffusion models as geometric projections that iteratively move corrupted inputs toward the clean image manifold. Based on the manifold assumption that image data resides on a low-dimensional smooth manifold, the paper integrates a learned distance function to the probability vector fields to guide image reconstruction and generation. The method shows superior performance compared to the latent diffusion model on image restoration and generation tasks.

**Strengths:**

- The paper introduces a new view of the diffusion model as a projection onto the manifold.

- Using a distance-based geometric approach and a kernel-based probabilistic model, the paper tries to make an interpretable link between them and attempts to make a unified framework.

- Detailed definitions on loss, architectures, and training settings have been provided in the Appendix.

**Weaknesses:**

- While the idea of viewing the diffusion model as a projection onto the manifold is interesting, I cannot find a theoretical explanation or demonstration that the iterative process approximates a projection. Also, Equations 11 and 12 are heuristic updates without any guarantee of convergence.

- The formulation is overly complex without clear benefit. Distance function, kernels, and autoencoders introduce considerable complexity, but I do not see why it should be explicitly better than exisiting diffusion models. Empirical results cannot be the justification as the datasets are too small and baselines are too weak.

- The experiments are limited to simple datasets: MNIST and SCUT-FBP5500 datasets. It does not show general applicability or scalability. Experiments on datasets with the scale of CIFAR-10 or LSUN would be recommended. Also, the compared baselines are too weak and naive to say the complex formulation of the method should be used.

- As the model introduces additional networks and iterative updates, it should require a comparison of computational complexity to diffusion models. Does learning distance functions and using it cost significantly?

- Ablation analysis on the main components, like the distance function or kernels, would make the claim of the paper stronger.

**Questions:**

Please address the questions raised in the weakness section.

---

> ### Author Response · Authors · 2025-11-21
> **Response to Reviewer UwHW**
>
> We thank the reviewer for the constructive review and for the useful remarks.
>
> 1. **Theoretical Convergence**
>
> The reviewer noted a lack of theoretical guarantee for the iterative projection.
>
> Eq. 12 and 13 represent the flow along the score (the gradient of the log-probability), which naturally moves points towards high-probability regions.
>
> We have added Appendix A.4, showing that the ‘flow’ equations (12) and (13) are Tweedie’s formulas for the corresponding probability functions, and we added a proof that the distance to the manifold decreases along the flow.
>
> 2. **Formulation Complexity vs. Benefit**
>
> The reviewer felt the formulation (Distance, Kernels, AE) is overly complex.
>
> Actually, the computational complexity is comparable to LDM. LDM learns an AE and a noise-prediction network ($\epsilon_t(x)$). Our formulation learns an AE and a distance function $D(x)$. The number of networks is identical.
>
> The benefit is the explicit geometric meaning: The distance function $D(x)$ is the geometric analog of LDM's $\epsilon_t(x)$, but because it represents a true geometric distance, it allows us to perform blind restoration without being tied to a specific noise schedule.
>
> 3. **Scalability**
>
> We have addressed this by adding experiments on the CelebA dataset (Figures 18-19  and Table 11 in Appendix D), demonstrating that LMPPM scales to complex facial features and produces high-quality reconstructions.
>
> 4. **Ablation Analysis**
>
> Regarding the request to omit the distance function or kernels: Since the distance function $D(x)$ is the functional analog of LDM's noise predictor $\epsilon_t(x)$, removing it would functionally break the model. However, we agree on the importance of isolating contributions. We have included an ablation study with the use of only the AE networks without using the distance D(x) in the inference phase. We compared the results of this flow for our coupled networks model to those obtained with a DAE. The coupling to the distance function improves immensely the  results even without the use of the distance function in the inference stage. The results are reported in table 1.

---

### Official Review · Reviewer_oVYE · 2025-10-30

**Soundness:** 2
**Presentation:** 3
**Contribution:** 1
**Rating:** 4
**Confidence:** 4

**Summary:**

The paper introduce a geometric picture framing VAE, GAN diffusion with respect to the data manifold, and they interpreted diffusion models as iterative manifold projection. Then they derive a model / objective (LMPPM) based on this idea, and and show some improved performance regarding clearing image degradation on some datasets.

**Strengths:**

### Strength

- The conceptual discussion about the manifold geometry in diffusion, VAE and GAN is interesting, and think about diffusion model in this geometric way is laudable.

**Weaknesses:**

### Weakness

- The new methods in Sec.4 is not very convincing and/or not super well framed in the literature. Specifically, I feel there are so many connection to existing diffusion models, energy based model etc., just by re-interpreting the entities.
    - e.g. the first term of the loss in eq. 13 learn the distance or energy instead of the score vector itself.
    - the 5th term is very similar to the denoising score matching objective if we parametrize score by denoisers [^3,^4] since $z_i^{shift}$ is basically the denoiser. which is enforcing the gradient of distance to be nicely aligned to score
    - In this regard, it seems the main innovation is that we have a distance function without explicit conditioning of noise scale. But seems [^5,^6] also discussed / discovered that the time / noise scale conditioning is not necessary.

[^5] Sun, Q., Jiang, Z., Zhao, H., & He, K. (2025). Is Noise Conditioning Necessary for Denoising Generative Models?.

[^6] Kadkhodaie, Z., Guth, F., Simoncelli, E. P., & Mallat, S. (2024). Generalization in diffusion models arises from geometry-adaptive harmonic representations. ICLR

- From the algorithm or the method itself, I cannot see a clear reason why the proposed MPPM or LMPPM method is better than LDM. is it the case than LDM needs a certain noise / time conditioning, thus if you input the wrong noise / time, it will not correctly denoise the image? but for LMPPM, you have no time conditioning, so you are more robust in that regard?
    - Currently the FID in Table 1 is very high for LDM and DAE, which is a bit concerning. I feel something is wrong in the implmentation of these baselines…
    - Elucidating why LMPPM is better via ablation / control experiment can largely improve the paper, and increase my evaluation of the paper.

**Questions:**

- in abstract why do the authors say “*The foundational premise of generative AI for images is the assumption that images are inherently low-dimensional objects embedded within a high dimensional space*”? Seems generative AI can still work if images are not low dimensional objects…. I agree with the assumption, but do not think it’s a foundational premise of generative AI.
- I feel the geometric view of diffusion models (Sec 3, Fig. 2) is definitely correct and worth noting, but it’s also not entirely new. Authors could mention very similar figures as in Fig1 [^1] Fig4 [^2]. e.g. the quantity noted in eq. 8 has name in many papers, i.e. ideal denoiser [^3,^2], and the relation between score and denoiser has been known as tweedie’s formula [^4]. $\hat{x}_{\text{MMSE}} = \mathbb{E}[u \mid x] = x + \sigma^2 \nabla_x \log P(x)$

[^1] Chen, D., Zhou, Z., Wang, C., Shen, C., & Lyu, S. (2024). On the trajectory regularity of ode-based diffusion sampling. ICML https://arxiv.org/abs/2405.11326

[^2] Wang, & Vastola, (2024). The unreasonable effectiveness of gaussian score approximation for diffusion models and its applications. TMLR https://arxiv.org/abs/2412.09726

[^3] Karras, T., Aittala, M., Aila, T., & Laine, S. (2022). Elucidating the design space of diffusion-based generative models. NeurIPS

[^4] Efron, B. (2011). Tweedie’s formula and selection bias. Journal of the American Statistical Association

- Eq.14 is also known as ideal denoiser with delta mixture distribution / empirical distribution [^2,^3].

- As the authors pointed out the Riemannian geometry of data manifold through the generator of GAN, VAE have been studied for a while, some reference could be added for this tradition [^7,^8,^9].

[^7] Shao, H., Kumar, A., & Fletcher, P. T. (2017). The riemannian geometry of deep generative models. *CVPR Workshops*

[^8] Wang, B., & Ponce, C. R. (2021). The geometry of deep generative image models and its applications. ICLR

[^9] Chadebec, C., & Allassonnière, S. (2022). A geometric perspective on variational autoencoders. NeurIPS

---

> ### Author Response · Authors · 2025-11-21
> **Response to Reviewer OVYE 1**
>
> We thank the reviewer for the feedback and for finding the geometric perspective interesting. We especially thank the reviewer for guiding us to frame our work in a wider context of current research.
>
> 1. **Novelty and Relation to Energy-Based Models (EBM)**
>
> The main idea of this work is to provide a geometric framework for generative models that enables the reinterpretation of various methods. Our research also represents a step forward in closing the gap between the probabilistic approach and the geometric one.
>
> Regarding EBMs: There are formal similarities, but also crucial differences. While one can frame Eq. 5 as an energy model where $E(x)=\mathcal{D}_{\mathcal{M}}^{2}(x)$, standard EBMs use energy as a "black-box" with some implicit assumed metric that connects the ambient points to the energy level. Our coupled distance model has an inherent metric that has a precise geometric meaning (distance to the manifold).
>
> Furthermore, once we move to the realistic description in Eq. 4 (probability distribution in the ambient space), we are in a very different context than standard EBMs. The only remaining similarity is that the score is inferred from the probability distribution. We have added a discussion on this wider context in the revised manuscript.
>
> 2. **Relation to Denoising Score Matching (DSM) and Tweedie’s Formula**
>
> The similarity to DSM is also partial. While both methods learn a vector field directing points in the ambient space to the manifold, the mechanisms differ.
>
> DSM typically relies on Gaussian noise addition, where "distance" is implicitly measured by noise levels/time steps. This notion of implicit distance is misleading and is not well-defined for degradations that are NOT Gaussian noise. Indeed, our experiments clearly show that our method handles non-Gaussian degradations (e.g., scribbles, missing pixels, etc.) much better than methods relying on noise-conditioning.
>
> The major difference is that we introduce the distance function, without explicit conditioning on the noise scale. As the reviewer noted, recent works (Sun et al. 2025, Kadkhodaie et al. 2024) discuss removing noise conditioning. Our work provides the geometric justification and implementation for this via the distance function. The distance function is inherently the time/noise scale needed.
>
>
>
> 3. **Comparison to Kadkhodaie et al. [6] (Ambient vs. Latent Integration)**
>
> Please note that while there are formal similarities with yours Ref [6] in the probability analysis, the essence of our work is geometric and computationally distinct.
>
> In Kadkhodaie et al., the probability in the ambient space is a convolution over the whole ambient space: $P(y) = \int g_\sigma(y-x)P_{clean}(x)dx$. The integration is over the ambient space and there is no way to sample the clean images in this space.
>
> In our formulation (Eq. 2 & 3), we strictly use the Manifold Assumption. The integration is over the manifold (parameterized by latent $z$):
>
> $$P(y) = \int P(y|G(z))P(G(z))\sqrt{g(z)}dz = \int P(y|G(z))P(z)dz$$
>
> This difference is fundamental. We integrate over the Latent Space. This allows us to sample effectively from the manifold of clean images (via $G(z)$).
>
> Moreover, since in Ref [6] they cannot sample effectively clean images,  they replace the exact term E[y|x] with a denoiser $\hat{x}=f_\theta(x)$.  The analog of this denoiser in our notations is $G(F(x))$ which is geometrically less accurate. We elucidate  in our work the geometric meaning of these two objects and claim that they are not necessarily close.
>
> 4. **Advantages over LDM (Why is it better?)**
>
> The Advantages over LDM is highlighted by two main aspects:
>
> - Robustness (No Time Conditioning): As noted, we are free from time/noise conditioning, making the model robust to unknown noise types and levels.
>
> - Deterministic Flow: We do not build our vector field stochastically. For better clarity, we exemplified the difference between our method to DM by a "Toy Example" (Figures 10-11). It shows that DM concentrates samples in dense regions of the data, whereas MPPM follows the geometric projection, leading to higher fidelity in low-density regions.
>
>
> 5. **Foundational Premise (Low-Dimensionality)**
>
>  We agree that generative AI can work without this assumption, but the fact that images are intrinsically low-dimensional is implicitly assumed in almost all modern methods (AE, VAE, GAN, LDM) that utilize a latent space with significantly lower dimensions than the pixel space. We have refined the phrasing in the abstract.

---

> > ### Author Response · Authors · 2025-11-21
> > **Response to Reviewer OVYE 2**
> >
> > 6. **Baseline Implementation (FID scores)**
> >
> > We acknowledge the FID scores for LDM are high. We used a standard LDM implementation to ensure a fair "out-of-the-box" comparison. However, the "Toy Example" (Figure 10-11) we added helps explain the performance gap: MPPM is geometrically more accurate in finding the closest manifold point, which is critical for restoration metrics (SSIM/L2), even if perceptual metrics vary. The concentration of DM around the more probable area explains the gap in the FID. For sanity, we added additional  FIDs calculations for weak degradation. See appendix E.

---

### Official Review · Reviewer_uPVp · 2025-10-30

**Soundness:** 4
**Presentation:** 4
**Contribution:** 3
**Rating:** 8
**Confidence:** 4

**Summary:**

This paper introduces the Manifold-Probabilistic Projection Model (MPPM) and its latent variant (LMPPM), which unify geometric and probabilistic interpretations of generative modeling. The method interprets diffusion models as iterative projections onto the manifold of “good” images, defined through a distance function and an associated kernel-based probability density. The authors derive this formulation rigorously from geometric principles, introduce both ambient-space and latent-space implementations, and connect the model to classical autoencoder architectures.
The paper is well-written, mathematically detailed, and offers a clear conceptual framework that bridges geometry and probability in generative modeling.

**Strengths:**

* Excellent clarity and presentation of the theoretical derivation.
* Well-organized narrative: the geometric intuition, probabilistic extension, and algorithmic details are all coherent and rigorous.
* The proposed framework provides an elegant deterministic alternative to diffusion sampling, supported by sound intuition.
* Clear and readable mathematical notation throughout.

**Weaknesses:**

**Experimental scope:**
* Despite the theoretical strength, experiments are limited to MNIST and SCUT-FBP5500, which are small and relatively trivial datasets. After such a solid theoretical development, this weak experimental section feels like a missed opportunity.
* Evaluations rely mainly on the Latent MPPM variant, and mostly in a reconstruction setting rather than true generation.
* While reconstruction is a valid demonstration, it is not the most relevant metric for generative models. The paper would be much stronger if generation quality were assessed on more challenging datasets such as ImageNet 64×64, CelebA-HQ, or CIFAR-10. Even a small-scale generation study (e.g., 32×32), if focused, would make the contribution more complete.

**Focus dilution:**
The inclusion of reconstruction experiments makes the paper feel slightly misaligned with its main message. A more focused evaluation of generation performance would better highlight the model’s strengths.

**Minor technical comments:**
* Missing citations for the Eikonal equation (line 145) and kernel density estimation (line 185).
* Line 216: the term “normalized gradient” seems redundant since $|| D_M (x) || = 1$ by the Eikonal equation.
* Equation (10): unclear why $G(z)$ appears outside the exponential.
* Line 55: the authors do not **propose** the manifold assumption but rather **assume** it.

**Questions:**

I have no questions other than asking the authors to perform more focused experiments, as indicated in the "Weakness" section.

---

> ### Author Response · Authors · 2025-11-21
> **Response to Reviewer uPVp**
>
> We thank the reviewer for the encouraging assessment and for highlighting the clarity and theoretical strength of our work. We also thank the reviewer for the useful remarks.
>
> 1. **Experimental Scope and Datasets (CelebA)**
>
> We agree that demonstrating performance on more complex datasets is crucial. In the revised manuscript, we have added experiments on the CelebA dataset (see Appendix D, Figures 18, 19 and the corresponding Table 11). These results demonstrate that LMPPM scales effectively to complex, real-world distributions.
>
> 2. **Generation vs. Restoration**
>
> We would like to remark on the distinction between the generation task and the restoration task.
>
> In generation, one starts from an image of noise, and without any specific context, landing on any point on the manifold of clean images is a success. For example, in MNIST, any final image that looks like a valid digit is acceptable.
>
> By contrast, in the restoration task, the algorithm must find in the manifold of the clean images  the exact point/image that was degraded. This is not an easier task than generation, especially for strongly degraded images. Note that for this class of strongly degraded images, our algorithm shows the most significant improvement over other methods.
>
> Furthermore, while MNIST classification is considered  a solved problem, generative modeling and inverse problems on this dataset are still not trivial and demand sophistication to capture the exact manifold structure, as shown in our comparisons.
>
> 3. **Blindness to Degradation**
>
> A crucial advantage of our framework is that it is "blind" to the type of degradation. It performs well on various corruptions (scribble, over-sharpening, missing pixels, etc.) without needing explicit training or inference adjustments for each type, unlike many baselines, which require specific training for specific noise types and levels.
>
> 4. **Technical Corrections**
>
> - **Missing Citations:** We have added the missing citations for the Eikonal equation and Kernel Density Estimation.
>
> - **Normalized Gradient:** You noted that $\|\nabla D\| = 1$ makes normalization redundant. This is absolutely correct theoretically if the network $D$ finds the exact solution. However, since the learned network is only an approximation, we normalized the direction vector $\nabla D$ to enforce a unit length, providing better control over the step sizes in the flow towards the manifold.
>
> - **Equation (10) / Exponential Derivative:** You asked why $G(z)$ appears outside the exponential. This arises from the gradient of the Gaussian term:
>     $$ \nabla_{x} \exp\left(-\frac{\|x-G(z)\|^{2}}{2\sigma^{2}}\right) = -\frac{1}{\sigma^{2}}(x-G(z)) \exp\left(-\frac{\|x-G(z)\|^{2}}{2\sigma^{2}}\right)$$
>     We have clarified this derivation in the text. See also in the detailed derivation in appendix A.2
>
> - **Manifold Assumption:** We corrected the phrasing to "assume" rather than "propose".

---

### Author Response · Authors · 2025-11-21
**General Response**

We thank the reviewers for their time and insightful comments. We are encouraged by the positive feedback on the theoretical rigor (**Reviewers uPVp, Vt93**) and the geometric intuition (**Reviewer OVYE**).

Based on your feedback, we have updated the manuscript significantly. The major changes include:

- **New Experiments on CelebA**: To address concerns regarding dataset scale (Reviewers uPVp, UwHW), we added restoration results on CelebA (Appendix D, Figs. 18-19and Table 11).

- **Comparison with DiffBIR (ECCV 2024)**: We added a comprehensive comparison against DiffBIR; See Appendix D1, Fig. 20 and Table 11. (Addressing Reviewer Vt93)

- **Extended Theory and Relation to Previous Works**:  We changed the presentation to highlight previous works like Kadkhodaie-etal2023 and Sun-etal2025 . We elaborate on the probability distribution that they put forward over the ambient space and explain why the Gaussian choice for the conditional probability does not limit the restoration to Gaussian noise. This choice leads to Blind Image Denoising (BID) as is illustrated in the experimental part. We add to this probabilistic viewpoint the Manifold assumption and show that this enables advancement in this approach leading to tractable computations and clarity of concepts. Throughout the theoretical part, we added similarities and differences with existing literature (Energy-Based Models, noise conditioning,  BID and more).  (Addressing Reviewers oVYE, Vt93 )

- **Theoretical Proofs**: We added a reference and the connection of our theory to Tweedie’s formula and a proof that the distance to the manifold decreases along the flow. (Addressing Reviewer UwHW )

- **Ablation study**:  We studied the effect of our new distance function via an ablation study. (Addressing Reviewer UwHW)

- **Toy Example Analysis**: We added a 1D manifold embedded in 3D (Figs. 10-11) to demonstrate the geometric superiority of MPPM in low-density regions (addressing Reviewers OVYE, Vt93).

 - **FID**:  We added several experiments to verify that our implementations of DAE and DM/LDM are correct. (Addressing Reviewer oVYE)

**The changes are highlighted in** $\textcolor{blue}{blue}$

---

### Meta-Review · Area_Chair_HJy4 · 2025-12-27

**Summary:**

This paper offers a unified geometric and probabilistic interpretation of diffusion-based image models by formulating a geometric framework coupled with a kernel-based probabilistic approach.  Under this perspective, diffusion processes are reinterpreted as projections onto a manifold representing high-quality images, shedding light on their underlying mechanisms.  Motivated by this view, the authors introduce a deterministic alternative, the Manifold Probability Projection Model (MPPM), which is consistently defined in both pixel and latent spaces.

This paper is highly theoretical, with the authors providing extensive theoretical derivations and results. However, several reviewers have raised concerns regarding the underlying assumptions and certain technical details. Most importantly, the experimental evaluation is relatively limited. Although additional baselines and datasets were included during the rebuttal, the experiments remain insufficient to fully support the theoretical claims. The authors are encouraged to carefully consider the reviewers’ suggestions and strengthen both the experimental validation and theoretical analysis by evaluating the method against stronger baselines and on larger-scale datasets.

**Reviewer Concerns:**

Limited experimental scope and evaluation: Experiments are restricted to small and relatively simple datasets. Although the author supplemented the experiments on the CelebA dataset during the rebuttal, it might not have been sufficient and had limited credibility. I encourage authors to conduct experiments on more large-scale datasets.

Weak  baselines: The baseline being compared is too weak, and the baselines provided in the paper is not specifically designed for noise scenarios. The authors supplemented DiffBIR in the rebuttal stage, but still lacked many baselines. This is the area that this paper needs to focus on improving.

Connection with previous study: The author discussed in detail in the rebuttal stage the differences between the proposed method and previous methods and theories. This convincing result can address the concerns of the reviewers.

Insufficient theoretical justification: Although interpreting diffusion as a projection onto a data manifold is intriguing, the paper lacks rigorous theoretical support or convergence guarantees, and some core assumptions (e.g., Gaussian noise perturbations) may not hold in more general settings.  I think the authors have given a very good response, which can solve the reviewers' concerns.

Missing complexity analysis and ablations: The method introduces additional components and iterative procedures without accompanying computational cost analysis or ablation studies, making it difficult to assess whether the increased complexity yields meaningful benefits. I think the authors' response in the rebuttal stage is sufficient to resolve the reviewers' concerns.

**Reviewer Scores:**

For Reviewer uPVp, his evaluation is positive enough, so he won't change the score.

For Reviewer oVYE, I think he put forward some very insightful opinions. Although the author responded, it might not be able to fully convince him. Therefore, there is a high probability that he will keep his score unchanged.

For Reviewer UwHW, the authors gave excellent responses to most of his reviews. Therefore, I think he might improve to 6.

For Reviewer Vt93, he had a heated discussion with the author. However, I think the author's response is not comprehensive, especially on the missing baseline. Therefore, I don't think he will change the score.

Overall, I believe that experimental shortcomings make this paper insufficient to meet the acceptance criteria. I encourage the author to add more experiments to improve the paper.

---

### Decision · Program_Chairs · 2026-01-26

Reject